# SyncTwin: Treatment Effect Estimation with Longitudinal Outcomes

**Zhaozhi Qian**
University of Cambridge
zq224@cam.ac.uk

**Yao Zhang**
University of Cambridge
yz555@cam.ac.uk

**Ioana Bica**
University of Oxford
The Alan Turing Institute
ioana.bica@eng.ox.ac.uk

**Angela Mary Wood**
University of Cambridge
amw79@medschl.cam.ac.uk

**Mihaela van der Schaar**
University of Cambridge
UCLA
The Alan Turing Institute
mv472@cam.ac.uk

## Abstract

Most of the medical observational studies estimate the causal treatment effects using electronic health records (EHR), where a patient's covariates and outcomes are both observed longitudinally. However, previous methods focus only on adjusting for the covariates while neglecting the temporal structure in the outcomes. To bridge the gap, this paper develops a new method, SyncTwin, that learns a patient-specific time-constant representation from the pre-treatment observations. SyncTwin issues counterfactual prediction of a target patient by constructing a synthetic twin that closely matches the target in representation. The reliability of the estimated treatment effect can be assessed by comparing the observed and synthetic pre-treatment outcomes. The medical experts can interpret the estimate by examining the most important contributing individuals to the synthetic twin. In the real-data experiment, SyncTwin successfully reproduced the findings of a randomized controlled clinical trial using observational data, which demonstrates its usability in the complex real-world EHR.

## 1   Introduction

Estimating the causal individual treatment effect (ITE) using observational data has become increasingly common in the medical literature due to the popularization of electronic health records (EHR). The EHR is a *longitudinal* collection of records: it contains repeated measurements of a patient's health condition over irregular time intervals. Although the treatments may also vary over time, many studies consider a *point treatment*, where the treatment allocation is performed at some observed time and stays fixed during the study [12, 44, 34, 5, 51]. Point treatment is widely applicable to problems involving one-off treatments (e.g. surgical operations) or treatments that do not change frequently (e.g. long term medication for chronic disease). We will refer to the setting above as Longitudinal and Irregularly sampled data with Point treatment setting, or *LIP*. The LIP setting will be the focus of this work (Figure 1 A).

The LIP setting is different from the conventional settings in causal inference. This is because we observe the *pre-treatment* outcome $y_t$ over time leading to the treatment allocation. These pre-treatment outcomes may unveil the inherent temporal structure in the outcome time series (e.g. trend and seasonality), leading to better ITE estimation over time. In contrast, the conventional settings

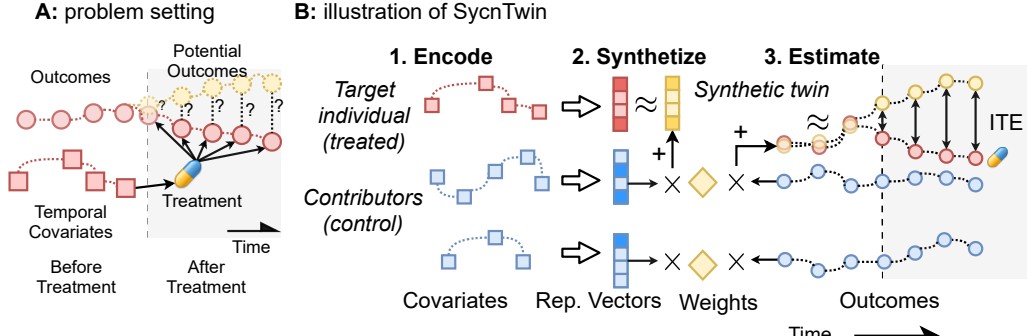

Figure 1: **A**: Illustration of the LIP setting. Yellow dots: the potential outcomes. **B**: Illustration of SyncTwin (shaded area: the time points after the treatment allocation). 1. Temporal covariates are encoded as representation vectors. 2. The synthetic twin of a treated target individual is constructed as the weighted average of the few contributors from the control group. 3. The difference between the observed outcome and the synthetic twin outcome estimates ITE. Similar procedures can be carried out for control or new individuals.

only consider the outcomes *after* treatment initiation [54, 52, 36, 43, 6]. This is true even for works that deal with dynamic treatment allocations [40, 35, 11].

A naive way to incorporate pre-treatment outcomes into the standard ITE methods is to treat them as additional covariates. This approach is viable but inadequate: the pre-treatment outcomes are arguably much more closely linked to the outcomes after treatment than the covariates — they hence deserve special considerations, e.g. modifying the architecture or loss function to reflect their importance. Without such modification, the ITE methods will not incorporate our prior belief on the importance of the pre-treatment outcomes, which may lead to worse performance.

To bridge this gap, we propose SyncTwin, a novel ITE estimation method tailored for the LIP setting. We assume that the temporal outcomes are generated by individualized latent factors and time-varying latent trends. This assumption is similar to the "factor model" assumption commonly used in Econometrics [1] and allows us to achieve the right balance between the parametric assumption and modeling flexibility. Figure 1 B illustrates the schematics of SyncTwin using an example with two treatment options (treated and control). SyncTwin first uncovers the individualized latent factors using representation learning. For a target individual, SyncTwin selects and weights a few contributors based on their latent factors and a sparsity constraint. It proceeds to construct a synthetic twin whose temporal outcomes are the weighted average of the contributors. Finally, the ITE is estimated as the difference between the target individual and the synthetic twin's outcomes.

Unlike conventional methods, SyncTwin does not use the learned latent factors to directly predict the outcomes; instead it uses these variables to find the contributors and their importance weights so as to construct the synthetic twin. This approach brings two bonus features; both are important to medical applications. (1) We can calculate an individualized estimation error bound based on the actual and the synthetic outcomes *before* treatment. In practice, the clinician can accept the recommended treatment when the error bound is below a threshold and resort to expert knowledge otherwise. (2) We can identify the most important contributors to an estimate as the ones who receive the highest weights. The clinician can further examine these contributors (e.g. whether they are indeed similar to the target individual) to validate and interpret the estimate.

**Contributions**. (1) We develop SyncTwin to leverage the temporal structure in the outcomes for better ITE estimation. (2) We provide theoretical justifications for each step involved in SyncTwin and prove an individualized error bound that can be used to identify untrustworthy estimates. (3) In addition to extensive simulations, we conduct an observational study using real EHR data and successfully reproduced the findings of a randomized controlled clinical trial with SyncTwin.

## 2 Problem setting

We consider an observational study with $N$ individuals indexed by $i \in [N] = \{1, \ldots, N\}$. Let $a_i \in \{0, 1\}$ be the treatment indicator with $a_i = 1$ if $i$ received the treatment at some time and $a_i = 0$

otherwise[1]. We realign the time steps such that all treatments were initiated at time $t = 0$. Let $\mathcal{I}_1 = \{i \in [N] \mid a_i = 1\}$ and $\mathcal{I}_0 = \{i \in [N] \mid a_i = 0\}$ be the set of the treated and the control individuals respectively. Denote $N_0 = |\mathcal{I}_0|$ and $N_1 = |\mathcal{I}_1|$ as the sizes of the groups.

Let $\mathbf{X}_i = [\mathbf{x}_{is}]_{s \in [S_i]}$ be the temporal covariates consisting of a sequence of observations $\mathbf{x}_{is} \in \mathbb{R}^D$. Let $\mathbf{M}_i = [\mathbf{m}_{is}]_{s \in [S_i]}$ be the sequence of binary masking vectors $\mathbf{m}_{is} \in \{0, 1\}^D$, where $[\mathbf{m}_{is}]_d = 1$ indicates the $d^{\text{th}}$ element of $\mathbf{x}_{is}$ is measured and $[\mathbf{m}_{is}]_d = 0$ otherwise. The entire sequence $\mathbf{X}_i$ contains $S_i \in \mathbb{N}$ observations taken before treatment at times $\mathcal{T}_i = [t_{is}]_{s \in [S_i]}$, $t_{is} < 0$. The maximum sequence length $S = \max_i(S_i)$.

The outcome $\mathbf{y}_{it} \in \mathbb{R}$ is observed at time $t \in \mathcal{T}^- \cup \mathcal{T}^+$. Let $\mathcal{T}^- = \{-M, \ldots, -1\}$ and $\mathcal{T}^+ = \{0, \ldots, H - 1\}$ be the observation times before and after treatment allocation. We arrange the outcomes after treatment into a $H$-dimensional vector denoted as $\mathbf{y}_i = [\mathbf{y}_{it}]_{t \in \mathcal{T}^+} \in \mathbb{R}^H$. Similarly define pre-treatment outcome vector $\mathbf{y}_i^- = [\mathbf{y}_{it}]_{t \in \mathcal{T}^-} \in \mathbb{R}^M$.

Using the potential outcome framework [41], let $\mathbf{y}_{it}(a_i) \in \mathbb{R}$ denote the potential outcome at time $t$ in a world where $i$ received the treatment as indicated by $a_i$. Let $\mathbf{y}_i(1) = [\mathbf{y}_{it}(1)]_{t \in \mathcal{T}^+} \in \mathbb{R}^H$, and $\mathbf{y}_i^-(1) = [\mathbf{y}_{it}(1)]_{t \in \mathcal{T}^-} \in \mathbb{R}^M$. Similarly, let $\mathbf{y}_i(0) = [\mathbf{y}_{it}(0)]_{t \in \mathcal{T}^+}$ and $\mathbf{y}_i^-(0) = [\mathbf{y}_{it}(0)]_{t \in \mathcal{T}^-}$. The individual treatment effect (ITE) is defined as $\tau_i = \mathbf{y}_i(1) - \mathbf{y}_i(0) \in \mathbb{R}^H$.

## 3 ITE estimation via SyncTwin

SyncTwin estimates the ITE by predicting both potential outcomes $\mathbf{y}_i(1)$ and $\mathbf{y}_i(0)$ and take the difference. Note that for an individual in the dataset $i \in [N]$, it is only necessary to predict the counterfactual outcome $\mathbf{y}_i(1 - a_i)$ because the factual outcome $\mathbf{y}_i(a_i) = \mathbf{y}_i$ is observed (under the consistency assumption to be discussed later). Without loss of generality, we will use estimating $\mathbf{y}_i(0)$ as an example in the following sections. One can estimate $\mathbf{y}_i(1)$ using the same method.

### 3.1 Assumptions

SyncTwin relies on three assumptions. **(1)** *Stable Unit Treatment Value Assumption* [41]: $\mathbf{y}_{it}(a_i) = \mathbf{y}_{it}$, $\forall i \in [N]$, $t \in \mathcal{T}^- \cup \mathcal{T}^+$. **(2)** *No anticipation*, also known as causal systems [4]: $\mathbf{y}_{it} = \mathbf{y}_{it}(1) = \mathbf{y}_{it}(0)$, $\forall t \in \mathcal{T}^-$, $i \in [N]$. **(3)** *Data generating process (DGP)*. The DGP assumption involves two parts. First, the causal directed acyclic graph (DAG) is specified in Figure 2 [37], which involves a latent factor $\mathbf{c}_i \in \mathbb{R}^K$. Secondly, we assume the potential outcomes have a parametric form:

$$\mathbf{y}_{it}(0) = \mathbf{q}_t^\top \mathbf{c}_i + \xi_{it}, \quad \forall t \in \mathcal{T}^- \cup \mathcal{T}^+, \tag{1}$$

where $\mathbf{q}_t \in \mathbb{R}^K$, $K < \min(M, H)$ is a weight vector and $\xi_{it}$ is the white noise. Equation 1 is commonly referred to as a *latent factor model* with $\mathbf{c}_i$ as the individualized latent factor and $\mathbf{q}_t$ as the time-varying latent trend [10]. Without loss of generality, we require the weight vectors $||\mathbf{q}_t|| = 1$, $\forall t \in \mathcal{T}^- \cup \mathcal{T}^+$ [50].

**Discussion.** SyncTwin differs from the nonparametric ITE estimation methods [27] because it assumes the parametric form in Equation 1. In essence, it means that we can fully separate the time effect from the individual effect. We would like to highlight two aspects of this assumption. First, the latent factor $\mathbf{c}_i$ does not change over time. It provides a constant link between the pre-treatment outcomes and the post-treatment outcomes. Secondly, the time-varying latent trend $\mathbf{q}_t$ is common to all individuals. Together, they ensure that any linear combination of different individuals' outcomes will automatically preserve $\mathbf{q}_t$, i.e. $\sum_i b_i \mathbf{y}_{it}(0) = \mathbf{q}_t^\top \sum_i b_i \mathbf{c}_i + \sum_i b_i \xi_{it}$, for any weights $b_i \in \mathbb{R}$, $\forall i \in [N]$. For this reason, SyncTwin

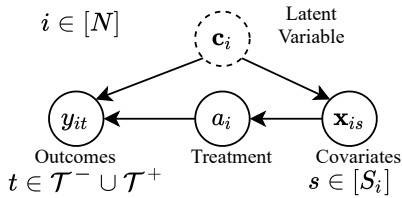

Figure 2: The DAG of the assumed data generating model.

bypasses $\mathbf{q}_t$ and directly finds the weights $b_i$'s to issue counterfactual predictions. As we will show later, doing so allows us to derive various theoretical results to inform the design of SyncTwin . It also leads to a checking procedure to validate estimation quality and improvements in interpretability.

---

[1]SyncTwin can model more than two treatment options, but we focus on binary treatments for illustration.

The DAG in Figure 2 is different from the one commonly used in the static setting [41]. We assume the latent factor $c_i$ captures the physiological factors that impact both the outcome and the covariates. Here, the covariates $x_{is}$ is a "confounder" in a general sense because it opens up a backdoor path from treatment to outcome, i.e. $a_i \rightarrow x_{is} \rightarrow c_i \rightarrow y_{is}$ [37]. Based on the backdoor criterion [37], adjusting for the covariates $x_{is}$ is sufficient to identify the treatment effect from observational data.

We compare our assumptions with those used in the related works in Appendix A.2. In particular, we show that our assumption is weaker than the one used by Synthetic Control, a widely-applied method in Econometrics [2]. In Appendix A.5, we further demonstrate the plausibility of our assumptions in real-world scenarios. In the simulation study (Section 5.1) we show experimentally that SyncTwin performs well even when the data is not generated from the assumed DGP exactly but instead from a set of differential equations. Finally, and perhaps most importantly, we show that based on our assumptions, SyncTwin is able to reproduce the findings of a large-scale randomized controlled trial using observational data (Section 5.2).

## 3.2 Learning to represent temporal covariates

The latent factor $c_i$ plays an important role as it affects the covariates $x_{is}$ and the outcomes $y_{it}$. SyncTwin uses deep neural networks to learn the representation $\tilde{c}_i$ as a proxy for $c_i$.

SyncTwin is agnostic to the exact choice of architecture as long as the network translates the irregularly sampled temporal covariates into a fixed-sized representation vector. For this reason, we use the well-proven sequence-to-sequence architecture (Seq2Seq) [47] with a standard attentive encoder [9] and a LSTM decoder [26]. The learned representation $\tilde{c}_i = f_e(X_i, M_i, \mathcal{T}_i; \theta_e)$, where $f_e$ is the encoder with trainable weights $\theta_e$. The reconstructed covariates $\hat{X}_i = f_d(\tilde{c}_i, \mathcal{T}_i; \theta_d)$, where $f_d$ is the decoder with trainable weights $\theta_d$. To ensure the learned representation $\tilde{c}_i$ is a *linear* predictor of the potential outcome $y_i(0)$ (Equation 1), we introduce a trainable parameter $\tilde{Q} \in \mathbb{R}^{H \times K}$ and define $\tilde{y}_i(0) := \tilde{Q} \cdot \tilde{c}_i$. Note that using a nonlinear function to map from $\tilde{c}_i$ to $\tilde{y}_i(0)$ will be *inconsistent* with the DGP and will not uncover the latent factor $c_i$ as desired.

We train the networks end-to-end by optimizing the loss function $\mathcal{L}^{tr} = \lambda_r \mathcal{L}_r + \lambda_p \mathcal{L}_s$, where $\lambda_r$ and $\lambda_p$ are hyper-parameters[2] balancing the supervised loss $\mathcal{L}_s$ and the reconstruction loss $\mathcal{L}_r$:

$$\mathcal{L}_s(\mathcal{D}_0) = \sum_{i \in \mathcal{D}_0} ||\tilde{y}_i(0) - y_i(0)||^2; \quad \mathcal{L}_r(\mathcal{D}_0, \mathcal{D}_1) = \sum_{i \in \mathcal{D}_0 \cup \mathcal{D}_1} ||(\hat{X}_i - X_i) \odot M_i||^2, \quad (2)$$

where $\mathcal{D}_0 \subseteq \mathcal{I}_0$ and $\mathcal{D}_1 \subseteq \mathcal{I}_1$ are training data, $m_{is}$ is the masking vector, $\odot$ represents element-wise product and $|| \cdot ||$ is the $L_2$ norm. In Proposition 1, we show that minimizing the supervised loss $\mathcal{L}_s$ will reduce the error bound on the learned representations, making them closer to the true latent factor $c_i$. The proof and the motivations for $\mathcal{L}_r$ are shown in A.1.1.

**Proposition 1** (Error bound on the learned representations). *Under the assumptions in Section 3.1, the total error on the learned representations for the control is bounded by:*

$$\sum_{j \in \mathcal{I}_0} ||c_j - \tilde{c}_j|| \leq \beta \mathcal{L}_s + \sum_{j \in \mathcal{I}_0} ||\xi_j||, \quad (3)$$

*where $\mathcal{L}_s$ is the supervised loss, $\beta$ is a constant depending on $q_t$ and $\tilde{Q}$, and $\xi_j$ is the white noise (Equation 1 and 2).*

## 3.3 Synthesizing the twin

At this point, one may be temped to predict the counterfactual outcome $y_i(0)$ of a treated individual $i \in \mathcal{I}_1$ with the output of the neural network $\tilde{y}_i(0)$. However, $\tilde{y}_i(0)$ is issued by a black-box neural network, which is not easily interpretable. As a remedy, SyncTwin explicitly constructs a synthetic twin who matches the target in representations. As we will show, this approach is able to control estimation bias and is more interpretable.

The synthetic twin of a target individual is defined by a set of weights $\boldsymbol{b}_i := [b_{ij}]_{j \in \mathcal{I}_0} \in \mathbb{R}^{N_0}$, each associated with a contributor in the control group $j \in \mathcal{I}_0$ (or treatment group when predicting $\hat{y}_i(1)$).

---

[2]The hyperparamter sensitivity is studied in A.13. $\lambda_r$ and $\lambda_p$ do not significantly impact the performance.

SyncTwin solves the following optimization problem to find the weights:

$$\boldsymbol{b}_i = \arg\min_{\tilde{\boldsymbol{b}}_i} \|\tilde{\mathbf{c}}_i - \sum_{j \in \mathcal{I}_0} \tilde{b}_{ij} \tilde{\mathbf{c}}_j\|^2 \quad \text{s.t. } \tilde{b}_{ij} \geq 0, \ \forall j \in \mathcal{I}_0 \text{ and } \sum_{j \in \mathcal{I}_0} \tilde{b}_{ij} = 1, \tag{4}$$

where $\tilde{\mathbf{c}}_j$ is the representation learned by the encoder. Denote the loss function in Equation 4 as $\mathcal{L}_m$. We will call $\mathcal{L}_m$ the *matching loss* because it indicates how well the synthetic twin matches the target individual in representations. Let $\mathrm{d}_i^c$ denote the optimal value of $\mathcal{L}_m$ and let $\hat{\mathbf{c}}_i$ be the synthetic representation given by the solution $\boldsymbol{b}_i$:

$$\mathrm{d}_i^c := \|\tilde{\mathbf{c}}_i - \sum_{j \in \mathcal{I}_0} b_{ij} \tilde{\mathbf{c}}_j\|^2; \quad \hat{\mathbf{c}}_i := \sum_{j \in \mathcal{I}_0} b_{ij} \tilde{\mathbf{c}}_j \tag{5}$$

The optimization procedure is detailed in Appendix A.8. SyncTwin predicts the potential outcome $\mathrm{y}_{it}(0)$ for a target individual $i$ using the same weights $\boldsymbol{b}_i$: $\forall t \in \mathcal{T}^- \cup \mathcal{T}^+$,

$$\hat{\mathrm{y}}_{it}(0) = \sum_{j \in \mathcal{I}_0} b_{ij} \mathrm{y}_{jt}(0) = \sum_{j \in \mathcal{I}_0} b_{ij} \mathrm{y}_{jt}, \tag{6}$$

where the last equality follows from the consistency assumption (Section 3.1). Denote $\hat{\mathbf{y}}_i(0) = [\hat{\mathrm{y}}_{it}(0)]_{t \in \mathcal{T}^+}$ as the predicted potential outcomes after treatment initiation.

The estimator in Equation 6 follows directly from the DGP assumption (Equation 1). To see this, remember that $\mathrm{y}_{it}(0)$ is *linear* in the latent factor $\mathbf{c}_i$. Hence, if we find a set of weights $b_{ij}^*$ to match the latent factor, i.e. $\mathbf{c}_i = \sum_j b_{ij}^* \mathbf{c}_j$, the same set of weights will also match the outcome $\mathrm{y}_{it}(0) \approx \sum_j b_{ij}^* \mathrm{y}_{jt}(0)$ up to random noise. In practice, since we do not observe $\mathbf{c}_i$, we have to perform matching on the learned representation $\tilde{\mathbf{c}}_i$ as in Equation 4.

Therefore we can see that the quality of the estimator $\hat{\mathrm{y}}_{it}(0)$ depends on two things: **(1)** whether the learned representation $\tilde{\mathbf{c}}$ is close to the true latent factor $\mathbf{c}$ and **(2)** whether good matching weights $\boldsymbol{b}_i$ can be found on the learned representation. SyncTwin attempts to fulfill the first condition by representation learning in Section 3.2. The second condition is facilitated by solving the optimization problem in Equation 4. Proposition 2 formalizes this intuition and shows that when a perfectly-matching twin is found, the estimation error only depends on the quality of the learned representations (proved in A.1.1).

**Proposition 2** (Bias bound on counterfactual prediction). *Suppose that $\mathrm{d}_i^c = 0$ for some $i \in \mathcal{I}_1$ (Equation 1 and 5), the absolute value of the counterfactual prediction bias for $i$ is bounded by:*

$$|\mathbb{E}[\hat{\mathbf{y}}_i(0) - \mathbf{y}_i(0)]| \leq |\mathcal{T}^+| \left( \|\mathbf{c}_i - \tilde{\mathbf{c}}_i\| + \sum_{j \in \mathcal{I}_0} \|\mathbf{c}_j - \tilde{\mathbf{c}}_j\| \right).$$

**Notes on Interpretability.** In addition to reducing bias by learning and matching representations, SyncTwin is also more interpretable. In particular, the weights $\boldsymbol{b}_i$ can be interpreted as the "contribution" or "importance" of a contributor $j$ to the target $i$ due to the optimization constraints. We can create a shortlist of the most important contributors based on $\boldsymbol{b}_i$. A domain expert can understand the rationale behind the estimate by examining the shortlist. For instance, they can check whether the important contributors share similar disease progression patterns as the target individual.

This procedure corresponds to the notion of *data point interpretability* or *example-based interpretability* [45], where one explains the prediction by presenting the most relevant data points to the users. Note that the learned representations $\tilde{\mathbf{c}}_i$ are internal to the algorithm; we do not intend to show these representations to the user or to make interpretations on them.

### 3.4 Calculating individualized error bound with pre-treatment outcomes

Since Equation 6 applies to all time points before and after treatment, let $\hat{\mathbf{y}}_i^-(0) = [\hat{\mathrm{y}}_{it}(0)]_{t \in \mathcal{T}^-}$ be the predicted *pre-treatment* potential outcome vector. We define $\mathrm{d}_i^y$ as the estimation error in the pre-treatment period:

$$\mathrm{d}_i^y = \|\hat{\mathbf{y}}_i^-(0) - \mathbf{y}_i^-(0)\|_1 = \|\hat{\mathbf{y}}_i^-(0) - \mathbf{y}_i^-\|_1, \tag{7}$$

where $\|\cdot\|_1$ is the vector $\ell_1$-norm. The second equality allows $\mathrm{d}_i^y$ to be evaluated for all individuals. It holds because of the no anticipation assumption (Section 3.1). Achieving a small error $\mathrm{d}_i^y$ implies

Table 1: **Problem settings considered in the literature**. "Static": observed (or allocated) only once; "Regular": observed (or allocated) over time at a regular frequency; "Irregular": observed over time irregularly; "-": not observed or modeled. * can be extended to Irregular. † can be extended to Regular. LIP: Longitudinal, Irregular, Point treatment.

| Setting | Example | Pre-treatment $\mathbf{X}$ | Pre-treatment $\mathbf{y}^-$ | Treatment $a$ | Post-treatment $\mathbf{y}$ | Nonlinear $f$: $\mathbf{y} = f(\mathbf{X})$ |
|---------|---------|-------------|-------------|-----------|----------------|-----------------|
| Static | [43] | Static* | - | Static | Static† | ✓ |
| DT | [11] | Regular* | - | Regular | Regular | ✓ |
| SC | [2] | Regular | Regular | Static | Regular | ✗ |
| LIP (This work) | This work | Irregular | Regular | Static | Regular | ✓ |

that the synthetic twin $\hat{\mathbf{c}}_i$ matches well with the *true* latent factor $\mathbf{c}_i$ due to the linearity between $\mathbf{y}_i^-(0)$ and $\mathbf{c}_i$ (Equation 1). Since $\mathbf{c}_i$ is assumed to be constant over time, the pre-treatment error can be used to assess the post-treatment error. We formalize this intuition in Proposition 3 and use $\mathrm{d}_i^y$ to control the error in the post-treatment period (proved in A.1.1).

**Proposition 3** (Error control under no hidden confounders). *Given any target error threshold $\delta > 0$, define the acceptance group of treated individuals as*

$$\mathcal{A}_\delta = \left\{ i \in \mathcal{I}_1 \mid \mathrm{d}_i^y \leq \delta |\mathcal{T}^-|/|\mathcal{T}^+| \right\}.$$

*Under the assumptions in Section 3.1, the post-treatment estimation error $|\mathbb{E}[\hat{\mathbf{y}}_i(0)] - \mathbb{E}[\mathbf{y}_i(0)]| \leq \delta$, $\forall i \in \mathcal{A}_\delta$.*

Proposition 3 shows that we can control the estimation error to be below a certain threshold $\delta$ by rejecting the estimate if its error $\mathrm{d}_i^y$ during the pre-treatment period is larger than $\delta |\mathcal{T}^-|/|\mathcal{T}^+|$. Alternatively, we can rank the estimation trustworthiness for the individuals based on $\mathrm{d}_i^y$. This is helpful when the user is willing to accept a percentage of estimations which are deemed most trustworthy.

### 3.5 Training, validation and inference

We perform model training, validation and inference (testing) on three disjoint datasets. In summary, we train the encoder and decoder on the training data using the loss functions described in Section 3.2. The validation data is then used to validate and tune the hyper-parameters of the encoder and decoder. Then, we solve the optimization problem in Section 3.3 to find the weights $\mathbf{b}_i$ for the target individuals in the testing data. Finally, we compute the potential outcome estimator (Section 3.3) and the individualized error bound (Section 3.4). The pseudocode is described in A.7.

## 4 Related work

**Synthetic Control**. Similar to SyncTwin, Synthetic control (SC) [1] and its extensions [8, 7] estimate ITE based on synthetic outcomes. However, SC needs to *flatten* the temporal covariates $[\mathbf{x}_{is}]_{s \in [S_i]}$ into a fixed-sized (high-dimensional) vector $\underline{\mathbf{x}}_i$ and use it to construct the twin. As a result, SC does not allow the covariates to be variable-length or sampled at irregular frequencies (Table 1); otherwise $\underline{\mathbf{x}}_i$'s dimensionality will vary across individuals. This severely limits SC's usability in medical observational studies because such sampling irregularities are common in EHR. Moreover, SC assumes $\mathrm{y}_{it}(0) = \mathbf{q}_t^\top \underline{\mathbf{x}}_i + \xi_{it}$, i.e. the flattened covariates $\underline{\mathbf{x}}_i$ *linearly* predicts $\mathrm{y}_{it}(0)$, which is a special case of SyncTwin's assumption (to be discussed later in Equation 1) and seems unlikely in many medical applications (e.g. body mass index does not linearly relate to blood pressure).

**Static nonparametric ITE estimation**. Most ITE estimation methods in the literature consider the static setting and do not pay special attention to the outcomes before treatment $\mathbf{y}_i^-$ — these outcomes are treated as standard covariates $\mathbf{X}_i$, if at all (Table 1). However, the pre-treatment outcomes $\mathbf{y}_i^-$ may encode certain temporal structure that will persist even on the outcomes $\mathbf{y}_i$ after treatment (e.g. trend and seasonality). Explicitly unveiling such temporal structure will be very beneficial to accurately estimating and validating the ITE after treatment. Moreover, these methods make no parametric assumption on the data generating model [27] and they learn to predict the two potential outcomes

using deep neural networks [43, 54, 22]. Despite the recent progress in interpretable machine learning, these methods are still in general less interpretable than the parametric models such as SyncTwin whose coefficients can be directly understood by an expert.

**ITE estimation for dynamic treatments**. Several works consider ITE estimation for dynamic treatments (DT) [40]. In the DT setting (Table 1), *multiple* treatment decisions are made over time for each individual. ITE estimation is much more challenging for DT because a prior treatment $a_{t-1}$ may influence the temporal covariates $x_t$, which may in turn confound a latter treatment $a_t$. This problem is known as *temporal confounding*. Marginal structural model [40] and related works [42, 35, 11] have been proposed to overcome temporal confounding. In comparison, our setting considers a point treatment and therefore do not suffer from temporal confounding. As a result, although the works in the DT setting are applicable, they might be over-complicated for our problem. In addition, our previous discussion about the inadequacy in leveraging pre-treatment outcomes also applies to DT methods.

**Works with similar terminology**. Several *unrelated* works in the literature use terms such as "twin", which might cause confusion. We discuss these works in Appendix A.4.

# 5 Experiments

## 5.1 Simulation Study

**Data generation.**[3] In this simulation study, we evaluate SyncTwin on the task of estimating the LDL cholesterol-lowering effect of statins, a common drug prescribed to hypercholesterolaemic patients. Following our convention, the individuals are enrolled at $t = 0$, the covariates are observed in $\mathcal{T} = [-S, 0)$, where $S \in \{15, 25, 45\}$, and the ITE is to be estimated in the period $\mathcal{T}^+ = [0, 4]$. We start by generating a covariate $k_t^{in}$ for from a mixture distribution:

$$k_{it}^{in} = \delta_i \mathbf{f}_{it} + (1 - \delta_i)\mathbf{g}_{it}; \quad \delta_i \overset{\text{iid}}{\sim} \text{Bern}(p_i) \tag{8}$$

where $\text{Bern}(p)$ is the Bernoulli distribution with success probability $p$ and $\mathbf{f}_{it}$, $\mathbf{g}_{it}$ are drawn from two different distributions (specified in A.10). To introduce confounding bias, we vary $p_i$ for the treated and the control: $p_i = p_0, \forall i \in \mathcal{I}_0$ and $p_i = 1, \forall i \in \mathcal{I}_1$. The constant $p_0$ controls the degree of confounding bias (smaller $p_0$, larger bias). We simulate the outcome $y_t$ using the widely adopted Pharmacology model in the literature [16, 53, 32]. It is obtained by solving the differential equation 9 and adding independent white noise $\epsilon \sim \text{N}(0, 0.1)$.

$$\dot{p}_t = k_t^{in} - k \cdot p_t; \quad \dot{d}_t = a_t - h \cdot d_t; \quad \dot{y}_t = k \cdot p_t - k d_t (d_t + d_{50})^{-1} \cdot y_t. \tag{9}$$

where $\dot{y}_t$ is the derivative of $y_t$ and $a_t$ is the indicator of statins treatment. The covariates includes $\mathbf{x}_t = \{k_t^{in}, y_t, p_t\}$. The interpretation of all constants involved are presented in Appendix A.10. Finally, we introduce irregular sampling by creating masks $\mathbf{m}_{it} \sim \text{Bern}(m)$, where probability $m \in \{0.3, 0.5, 0.7, 1\}$. It is worth highlighting that the simulation data are not generated from SyncTwin's assumed DGP (1) but from domain-specific ODEs (9).

**Benchmarks**. From the Synthetic Control literature, we considered the original Synthetic Control method (SC) [2], Robust Synthetic Control (RSC) [7] and MC-NNM [8]. From the dynamic treatment literature, we compared against Counterfactual Recurrent Network (CRN) [11] and Recurrent Marginal Structural Network (RMSN) [35]. Note that RMSN belongs to the family of Marginal Structural Models (MSMs) and we use it as a representative of MSMs. We included a modified CFRNet, which was originally developed for the static setting [43]. We replaced its fully-connected encoder with the encoder used by SyncTwin to to allow CFRNet to model temporal covariates (Section 3.2). We also included a benchmark adapted from the counterfactual Gaussian Process (CGP) [42] in order to adjust for the patient level covariates. We use One-nearest Neighbour Matching (1NN) [46] as a baseline. The implementation details of all benchmarks are available in Appendix A.9. We compared with two ablated versions of SyncTwin. SyncTwin-$\mathcal{L}_r$ is trained only with reconstruction loss and SyncTwin-$\mathcal{L}_s$ only with supervised loss.

---

[3]The implementation of SyncTwin and the experiment code are available at `https://github.com/ZhaozhiQIAN/SyncTwin-NeurIPS-2021` or `https://github.com/orgs/vanderschaarlab/repositories`

Table 2: Mean absolute error on ITE under different levels of confounding bias $p_0$. $m = 1$ and $S = 25$ are used. Estimated standard deviations are shown in the parentheses. The best performer is in bold. Additional quantitative results are shown in Appendix A.11

| Method | $N_0 = 200$ | | | $N_0 = 1000$ | | |
| --- | --- | --- | --- | --- | --- | --- |
| | $p_0 = 0.1$ | $p_0 = 0.25$ | $p_0 = 0.5$ | $p_0 = 0.1$ | $p_0 = 0.25$ | $p_0 = 0.5$ |
| SyncTwin-Full | **0.323 (.039)** | **0.144 (.013)** | **0.128 (.008)** | 0.178 (.012) | **0.106 (.006)** | **0.094 (.005)** |
| SyncTwin-$\mathcal{L}_r$ | 0.353 (.040) | 0.171 (.016) | 0.135 (.010) | 0.256 (.026) | 0.146 (.013) | 0.102 (.006) |
| SyncTwin-$\mathcal{L}_s$ | 0.336 (.040) | 0.171 (.015) | **0.119 (.008)** | **0.145 (.013)** | 0.114 (.007) | 0.127 (.010) |
| SC | 0.341 (.041) | 0.151 (.024) | 0.149 (.018) | 0.258 (.050) | 0.166 (.034) | 0.214 (.036) |
| RSC | 0.842 (.045) | 0.361 (.020) | 0.322 (.019) | 0.310 (.016) | 0.298 (.014) | 0.302 (.014) |
| MC-NNM | 1.160 (.060) | 0.612 (.031) | 0.226 (.011) | 0.527 (.029) | 0.159 (.008) | 0.124 (.007) |
| CFRNet | 0.903 (.077) | 0.387 (.035) | 0.291 (.003) | 0.399 (.038) | 0.178 (.013) | 0.104 (.007) |
| CRN | 0.809 (.050) | 0.613 (.039) | 0.335 (.023) | 0.779 (.041) | 0.589 (.040) | 0.563 (.035) |
| RMSN | 0.418 (.032) | 0.391 (.029) | 0.334 (.027) | 0.478 (.039) | 0.414 (.034) | 0.390 (.032) |
| CGP | 0.660 (.043) | 0.610 (.039) | 0.561 (.035) | 0.826 (.056) | 0.693 (.047) | 0.602 (.038) |
| 1NN | 1.866 (.099) | 1.721 (.091) | 1.614 (.078) | 2.446 (.131) | 1.746 (.106) | 1.384 (.083) |

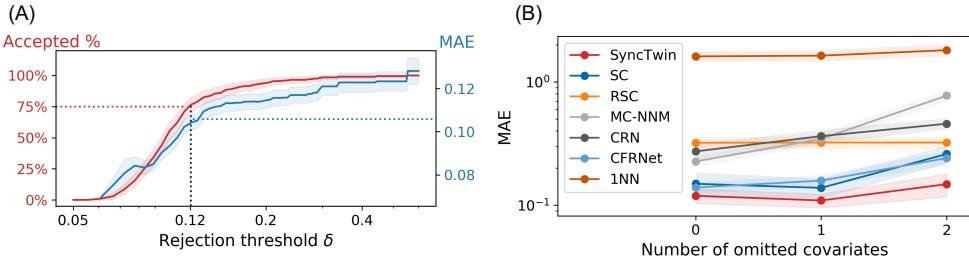

Figure 3: **(A)** The percentage of accepted estimates (red) and their MAE (blue) using various rejection thresholds $\delta$ on $\mathrm{d}_i^y$. Eg. setting $\delta = 0.12$ gives 75% acceptance rate and MAE around 0.105. **(B)** Performance when certain covariates are omitted (hidden confounders). Shaded area: 95% confidence interval.

**Evaluation metric**. In many medical studies, the outcome of interest is measured with heavy-tailed non-Gaussian noise [30]. In these settings, the mean absolute error (MAE) is preferred over the mean squared error (MSE) as an evaluation metric due to its robustness to the outliers [38]. For this reason, we will evaluate the mean absolute error (MAE) on counterfactual prediction: $\frac{1}{N_1} \sum_{i=1}^{N_1} ||\mathbf{y}_i(0) - \hat{\mathbf{y}}_i(0)||_1$. Note that MAE is also directly comparable to the error bound in Proposition 3 (i.e. the L1 norm).

**Main results**. Table 2 presents the results for various levels of confounding bias $p_0$. Additional results for different sequence length $S$ and sampling irregularity $m$ are shown in the tables in Appendix A.11. SyncTwin achieves the best or equally-best performance in all cases despite the assumed DGP does not exactly hold. The full SyncTwin with both loss functions also consistently outperforms the versions trained only with $\mathcal{L}_r$ or $\mathcal{L}_s$. In comparison, SC, RSC and MC-NNM underperform because their assumption that the flattened covariates $\underline{\mathbf{x}}_i$ *linearly* predict the outcome is violated (Section 4).

**Error bound**. SyncTwin allows the user to reject untrustworthy estimates based on $\mathrm{d}_i^y$. In Figure 3 (A) below, we show the scenarios of using various rejection thresholds on $\mathrm{d}_i^y$. As expected from Proposition 3, when the user increases the threshold, more estimates will be accepted but they will have a higher MAE. The fact that the two curves in the figure share the similar increasing trend suggests $\mathrm{d}_i^y$ is a good indicator of individual error: rejection based on a non-informative criterion will lead to a flat line of MAE. In practice, the user can use Figure 3 (A) to calibrate the threshold in order to achieve balance between the accuracy and the workload. For instance, if the user would like SyncTwin to accept 75% of the estimate, he or she should set threshold $\delta = 0.12$ according to the pre-treatment error $d_i^y$ and expect a MAE of around 0.105 for the post-treatment outcome.

**Sensitivity to omitted covariates**. In Figure 3 (B), we show the situation when some covariates are omitted from the analysis, thus making it harder to adjust for the confounding bias. In particular,

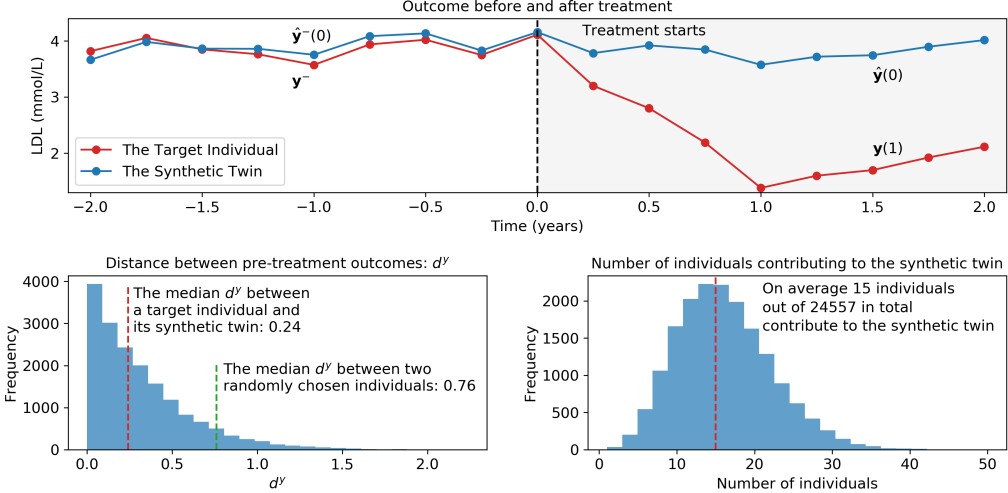

Figure 4: Top: the outcomes (LDL) before and after treatment of a target individual and its synthetic twin. Bottom left: histogram of distance $d^y$ (Equation 7). Bottom right: histogram of number of contributors used to construct the synthetic twin.

we set $\mathbf{x}_t = \{k_t^{in}, y_t, p_t\}$, $\mathbf{x}_t = \{y_t, p_t\}$ or $\mathbf{x}_t = \{y_t\}$ (omitting 0,1,2 covariates). We observe that SyncTwin is still the best-performing model when some covariates are omitted.

## 5.2 Experiment on real data

Few existing works in the literature validate the method by conducting a real-world observational study to replicate the findings of a randomized controlled trial (RCT), which is the gold standard of causal inference. Most existing works rely on synthetic or semi-synthetic data for validation, which may significantly under-represent the complexity of the real-world data.

To validate the usability of SyncTwin in real-world medical problems, we conduct an observational study to emulate a large-scale RCT—Heart Protection Study (HPS) [21, 20]. Although most existing methods have been validated in synthetic or semi-synthetic experiments [43, 11], few of them have been demonstrated to successfully reproduce the findings of a RCT from observational data. Since RCT is the gold standard of treatment effect estimation, the ability to reproduce a RCT provides strong evidence about the method's usability in medical research and facilitates the method's adoption in the medical community.

The HPS was conducted to investigate the treatment effect of statins, a drug commonly used to lower the LDL Cholesterol (LDL). It reported an a change of -1.26 mmol/L (SD=0.06) in LDL for participants randomised to statins versus placebo at the end of the first year after treatment [39]. The study enrolled 20,536 individuals and lasted for eight years, making it one of the largest studies to investigate the treatment effect of statins.

**Data Source**. We used medical records from English National Health Service general practices that contributed anonymised primary care electronic health records to the Clinical Practice Research Datalink (CPRD), covering approximately 6.9 percent of the UK population [25]. CPRD was linked to secondary care admissions from Hospital Episode Statistics, and national mortality records from the Office for National Statistics. We defined treatment initiation as the date of first CPRD prescription and the outcome of interest was the measured LDL. Known risk factors for LDL were selected as temporal covariates measured before treatment initiation: HDL Cholesterol, Systolic Blood Pressure, Diastolic Blood Pressure, Body Mass Index, Pulse, Creatinine, Triglycerides and smoking status. Our analysis is based on a subset of 125,784 individuals who met the enrollment criterion of HPS (Appendix A.15). They were split into three equally-sized subsets for training, validation and testing, each with 17,371 treated and 24,557 controls.

**Main findings**. SyncTwin estimates the average treatment effect as **-1.25** mmol/L (SD 0.01) by averaging the estimated ITE on testing data $\sum_{i \in \mathcal{D}^{te}} \hat{\tau}_{it}/|\mathcal{D}^{te}|$. The estimate is very close to the result reported in HPS, -1.26 mmol/L (SD=0.06), after taking into account the standard errors. In contrast, CRN and RMSN estimate the effect to be **-0.72** mmol/L (SD 0.01) and **-0.83** mmol/L (SD 0.01) respectively, which are significantly smaller than the trial's finding. Other benchmark methods (e.g. SC, RSC and MC-NNM) either cannot handle irregularly-measured covariates or do not scale to the size of the dataset. Our result suggests SyncTwin is able to overcome the confounding bias in the complex real-world datasets. Note that SyncTwin is also able to estimate ITE and the effect over different time horizons. However, since these results are not reported in HPS, we are not able to quantitatively evaluate them.

**Qualitative evaluation of ITE over time**. For each individual, we can visualize the outcomes before treatment and compare them with the synthetic twin to sense-check the estimate. The individual shown in Figure 4 (top) has a sensible ITE estimate because the synthetic twin matches its pre-treatment outcomes closely. In addition to visualization, we can calculate the individualized error bound based on $d_i^y$ (Equation 7). From Figure 4 (bottom left) we can see in most cases $d_i^y$ is small with a median of 0.24 mmol/L (compared to the average between any two randomly chosen individuals 0.76 mmol/L). This means if the expert can only tolerate an error of 0.24 mmol/L on ITE estimation, half of the estimates (those with $d_i^y \leq 0.24$ mmol/L) can be accepted (Section 3). As shown in Figure 4 (bottom right) on average only 15 (out of 24,557) individuals contribute to the synthetic twin ($b_{ij} > 0.01$). The medical experts can check these few contributors to interpret the estimate.

# 6 Conclusion and future work

We present SyncTwin, an ITE estimation method tailored for temporal outcomes with point treatment. SyncTwin achieves interpretability and strong performance in both simulated and real data experiments. In future works, we plan to extend SyncTwin to the dynamic treatment setting and to model the outcomes in continuous time. Developing additional assumptions to guarantee the robustness to certain types of unobserved confounders is also an interesting avenue for future research.

## Acknowledgments and Disclosure of Funding

We thank anonymous reviewers as well as members of the vanderschaar-lab for many insightful comments and suggestions. This work is supported by the US Office of Naval Research (ONR), the National Science Foundation (NSF Grant number:1722516), the Alan Turing Institute (under the EPSRC grant EP/N510129/1), and GlaxoSmithKline (GSK).

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
