# A  Appendix

**Summary of Appendices.**

Theoretical results.

- A.1: Proofs and discussions about all propositions.

Related works.

- A.2: Comparison of the causal assumptions
- A.3: Comparison of allowed temporal covariates
- A.4: Unrelated works with similar terminology

The SyncTwin algorithm.

- A.5: The generality of SyncTwin's assumed DGP
- A.6: Estimation for control and new individuals
- A.7: Algorithmic details and pseudocode
- A.8: Optimization for the matching loss $\mathcal{L}_m$

Simulation study.

- A.9: Implementation details of the benchmarks
- A.10: Details about the simulation model
- A.11: Additional simulation results
- A.12: Sparsity of weight $\mathbf{b}_i$
- A.13: Sensitivity of Hyper-Parameters
- A.14: Computation time

Experiment on real data.

- A.15: Additional results and statistics
- A.16: Cohort selection diagram in the CPRD study

**Theoretical results**

## A.1 Theoretical results

### A.1.1 Proofs of the propositions

**Proposition 1** Error bound on the learned representations. Given the assumptions in Section 3.1, the total error on the learned representations for the control [4] is bounded as follows:

$$\sum_{j \in \mathcal{I}_0} \|\mathbf{c}_j - \tilde{\mathbf{c}}_j\| \leq \beta \mathcal{L}_s + \sum_{j \in \mathcal{I}_0} \|\xi_j\|, \tag{10}$$

where $\mathcal{L}_s$ is the supervised loss in Equation 2 and $\xi_j$ is the white noise in Equation 1.

*Proof.* We start the proof from the definition of the supervised loss.

$$\begin{aligned}
\mathcal{L}_s &= \sum_{j \in \mathcal{I}_0} \|\tilde{\mathbf{Q}}\tilde{\mathbf{c}}_j - \mathbf{y}_j(0)\| \\
&= \sum_{j \in \mathcal{I}_0} \|\tilde{\mathbf{Q}}\tilde{\mathbf{c}}_j - (\mathbf{Q}\mathbf{c}_j + \xi_j)\| \\
&\geq \sum_{j \in \mathcal{I}_0} \left( \sum_{t \in \mathcal{T}^-} [\tilde{\mathbf{c}}_j^\top, -\mathbf{c}_j^\top] \begin{bmatrix} \tilde{\mathbf{q}}_t \\ \mathbf{q}_t \end{bmatrix} [\tilde{\mathbf{q}}_t^\top, \mathbf{q}_t^\top] \begin{bmatrix} \tilde{\mathbf{c}}_j \\ -\mathbf{c}_j \end{bmatrix} \right)^{\frac{1}{2}} - \\
&\quad \sum_{j \in \mathcal{I}_0} \|\xi_j\|^2 \\
&\geq \tilde{\beta}\sqrt{|\mathcal{T}^-|} \sum_{j \in \mathcal{I}_0} \|\tilde{\mathbf{c}}_j - \mathbf{c}_j\| - \sum_{j \in \mathcal{I}_0} \|\xi_j\|
\end{aligned} \tag{11}$$

where $\tilde{\beta}$ denotes the square root of the element of the matrices $\begin{bmatrix} \tilde{\mathbf{q}}_t \\ \mathbf{q}_t \end{bmatrix} [\tilde{\mathbf{q}}_t^\top, \mathbf{q}_t^\top], \forall t \in \mathcal{T}_-$, with the smallest absolute value. The first and second equations follow from Equation 2 and 1. Let $\beta$ denotes the constant $1/(\beta\sqrt{|\mathcal{T}^-|})$. Arranging the terms in inequality 11 and we prove Proposition 1. □

**Justification for the supervised loss**. Proposition 1 provide a justification for the supervised loss $\mathcal{L}_s$. By optimizing the supervised loss, SyncTwin learns the representation $\tilde{\mathbf{c}}_i$ that is close to the latent factor $\mathbf{c}_i$, which also reduces the bias bound on ITE in Proposition 2.

**Rationale for the reconstruction loss**. Although the bias bounds we developed so far do not include the reconstruction loss $\mathcal{L}_c$, we believe it is useful in real applications. Our reasoning follows from the fact that unsupervised or semi-supervised losses often improve the performance of deep neural networks [15, 14, 24]. In addition, the reconstruction loss ensures the representation $\tilde{\mathbf{c}}$ retains the information from the temporal covariates as required in the DAG (Figure 1). In our simulations (Section 5.1), we found that ablating the reconstruction loss leads to consistently worse performance (though the magnitude is somewhat marginal).

**Proposition 2** Bias bound on counterfactual prediction. Given the assumptions in Section 3.1 and suppose that $d_i^c = 0$ for some $i \in \mathcal{I}_1$ ($d_i^c$ is defined in Equation 5), the absolute value of the expected difference in the true and estimated potential outcome of $i$ is bounded by: (the expectations are taken with respect to the random noise $\xi_{it}$ in Equation 1)

$$\begin{aligned}
|\mathbb{E}[\hat{\mathbf{y}}_i(0)] - \mathbb{E}[\mathbf{y}_i(0)]| &\leq |\mathcal{T}^+| \| \sum_{j \in \mathcal{I}_0} b_{ij}\mathbf{c}_j - \mathbf{c}_i\| \\
&\leq |\mathcal{T}^+| \Big( \sum_{j \in \mathcal{I}_0} \|\mathbf{c}_j - \tilde{\mathbf{c}}_j\| + \|\mathbf{c}_i - \tilde{\mathbf{c}}_i\| \Big).
\end{aligned} \tag{12}$$

---

[4]This is the first term in the upper bound of the absolute value of the expected error in counterfactual prediction (R.H.S of Equation 12 Proposition 2)

*Proof.* We start the proof by observing

$$
\begin{aligned}
|\mathbb{E}[\hat{\mathbf{y}}_i(0)] - \mathbb{E}[\mathbf{y}_i(0)]| &= \sum_{t \in \mathcal{T}^+} |\mathbb{E}[\hat{\mathbf{y}}_{it}(0)] - \mathbb{E}[\mathbf{y}_{it}(0)]| \\
&= \sum_{t \in \mathcal{T}^+} |\mathbf{q}_t^\top (\sum_{j \in \mathcal{I}_0} b_{ij}\mathbf{c}_j - \mathbf{c}_i)| \\
&\leq \sum_{t \in \mathcal{T}^+} ||\mathbf{q}_t|| \cdot ||\sum_{j \in \mathcal{I}_0} b_{ij}\mathbf{c}_j - \mathbf{c}_i|| \\
&= |\mathcal{T}^+| \cdot ||\sum_{j \in \mathcal{I}_0} b_{ij}\mathbf{c}_j - \mathbf{c}_i||
\end{aligned}
\tag{13}
$$

where the first equation follows from the definition of ITE in Section 3. The second equation follows from Equation 1 and 6. The third line follows from Cauchy–Schwarz inequality. The fourth line uses the fact that $||\mathbf{q}_t|| = 1$. By definition, $\mathrm{d}_i^c = 0$ implies $\sum b_{ij}\tilde{\mathbf{c}}_j = \tilde{\mathbf{c}}_i$. Continuing the proof,

$$
\begin{aligned}
||\sum_{j \in \mathcal{I}_0} b_{ij}\mathbf{c}_j - \mathbf{c}_i|| &= ||\sum_{j \in \mathcal{I}_0} b_{ij}(\mathbf{c}_j - \tilde{\mathbf{c}}_j) - (\mathbf{c}_i - \tilde{\mathbf{c}}_i)|| \\
&\leq \sum_{j \in \mathcal{I}_0} b_{ij}||\mathbf{c}_j - \tilde{\mathbf{c}}_j|| + ||\mathbf{c}_i - \tilde{\mathbf{c}}_i|| \\
&\leq \sum_{j \in \mathcal{I}_0} ||\mathbf{c}_j - \tilde{\mathbf{c}}_j|| + ||\mathbf{c}_i - \tilde{\mathbf{c}}_i||,
\end{aligned}
\tag{14}
$$

where the second line follows from the triangular inequality and the third line relies on $\sum_{j \in \mathcal{I}_0} b_{ij} = 1$ and $b_{ij} \geq 0, \forall j \in \mathcal{I}_0$. Combining inequality 13 and 14, we prove the inequalities in Equation 12. $\square$

**Justification for the matching loss and** $\mathrm{d}_i^c$. Proposition 2 presents a justification for minimizing $\mathrm{d}_i^c$ (or the matching loss $\mathcal{L}_m$). Essentially, when the synthetic representations are matched with the target ($\mathrm{d}_i^c = 0$), the bias in ITE estimate is controlled by how close the learned representations $\tilde{\mathbf{c}}$ is to the true latent factor $\mathbf{c}$.

**Proposition 3** Error control under no hidden confounders. Suppose that all the outcomes are generated by the model in Equation 1, and that we reject the estimate $\hat{\tau}_i$ if the pre-treatment error $\mathrm{d}_i^y$ on $\mathcal{T}^-$ is larger than $\delta|\mathcal{T}^-|/|\mathcal{T}^+|$, the post-treatment ITE estimation error on $\mathcal{T}^+$ is below $\delta$.

*Proof.* Let $\mathbf{Q}^- = [\mathbf{q}_t]_{t \in \mathcal{T}^-}$ and $\mathbf{Q} = [\mathbf{q}_t]_{t \in \mathcal{T}^+}$ denote the matrix that stacks all the weight vectors $\mathbf{q}$'s before and after treatment as rows respectively where each $\mathbf{q}_t$ satisfies that $||\mathbf{q}_t|| = 1$ in Equation 1. The error $\mathrm{d}_i^y$ in Equation 7 can be decomposed into a representation error and a white noise error,

$$
\begin{aligned}
\mathrm{d}_i^y &= ||\hat{\mathbf{y}}_i^- - \mathbf{y}_i^-||_1 \\
&= ||\sum_{j \in \mathcal{I}_0} b_{ij}\mathbf{y}_j^- - \mathbf{y}_i^-||_1 \\
&= ||\sum_{j \in \mathcal{I}_0} b_{ij}(\mathbf{Q}^-\mathbf{c}_j + \xi_j) - (\mathbf{Q}^-\mathbf{c}_i + \xi_i)||_1 \\
&= ||\mathbf{Q}^-(\sum_{j \in \mathcal{I}_0} b_{ij}\mathbf{c}_j - \mathbf{c}_i)||_1 + ||\sum_{j \in \mathcal{I}_0} b_{ij}\xi_j - \xi_i||_1 \\
&\leq \sum_{t \in \mathcal{T}^-} ||\mathbf{q}_t|| ||\sum_{j \in \mathcal{I}_0} b_{ij}\mathbf{c}_j - \mathbf{c}_i|| + ||\sum_{j \in \mathcal{I}_0} b_{ij}\xi_j - \xi_i||_1 \\
&\leq |\mathcal{T}^-| ||\sum_{j \in \mathcal{I}_0} b_{ij}\mathbf{c}_j - \mathbf{c}_i|| + ||\sum_{j \in \mathcal{I}_0} b_{ij}\xi_j - \xi_i||
\end{aligned}
\tag{15}
$$

We can not estimate the error from the representation and white noise on the last line of Equation 15. Conservatively, we can say the representation error itself is larger or equal to $\mathrm{d}_i^y$ such that

$$
|\mathcal{T}^-| ||\sum_{j \in \mathcal{I}_0} b_{ij}\mathbf{c}_j - \mathbf{c}_i|| \geq \mathrm{d}_i^y,
$$

i.e.,
$$\| \sum_{j\in\mathcal{I}_0} b_{ij}\mathbf{c}_j - \mathbf{c}_i \| \geq d_i^y/|\mathcal{T}^-|. \tag{16}$$

The post-treatment error is upper bounded as follows,
$$\begin{aligned}
|\mathbb{E}[\hat{\mathbf{y}}_i(0)] - \mathbb{E}[\mathbf{y}_i(0)]| &= |\mathbb{E}[\hat{\mathbf{y}}_{it}(0)] - \mathbb{E}[\mathbf{y}_{it}(0)]| \\
&= \sum_{t\in\mathcal{T}^+} |\mathbf{q}_t^\top (\sum_{j\in\mathcal{I}_0} b_{ij}\mathbf{c}_j - \mathbf{c}_i)| \\
&\leq |\mathcal{T}^+| \| \sum_{j\in\mathcal{I}_0} b_{ij}\mathbf{c}_j - \mathbf{c}_i \| \\
&:= \sup_{\hat{\tau}_i} |\mathbb{E}[\hat{\tau}_i] - \mathbb{E}[\tau_i]|.
\end{aligned}$$

Using Equation (16), we have
$$\sup_{\hat{\tau}_i} |\mathbb{E}[\hat{\mathbf{y}}_i(0)] - \mathbb{E}[\mathbf{y}_i(0)]| \geq d_i^y|\mathcal{T}^+|/|\mathcal{T}^-|.$$

Conservatively, we reject the estimate $\hat{\tau}_i$ if $\sup_{\hat{\tau}_i} |\mathbb{E}[\hat{\tau}_i] - \mathbb{E}[\tau_i]|$ is larger than $\delta$. That is when
$$d_i^y > \delta|\mathcal{T}^-|/|\mathcal{T}^+|.$$

$\square$

**Why does $d_i^y$ indicate the trustworthiness of the estimation?** Proposition 3 shows that we can control the estimation error to be below a certain threshold $\delta$ by rejecting the estimate if its error $d_i^y$ during the pre-treatment period is larger than $\delta|\mathcal{T}^-|/|\mathcal{T}^+|$. Alternatively, we can rank the estimation trustworthiness for the individuals based on $d_i^y$ alone. This is helpful when the user is willing to accept a percentage of estimations which are deemed most trustworthy. We note that this proposition only holds under the assumption that the outcomes over time are generated by the model stated in Equation 1. The outcomes generated by such a model can be nonlinear and complicated due to the representation. However, the model assumes that the outcomes over time are linear functions of the same representation. This is the reason why the pre-treatment error can be used to assess the post-treatment error. We parameterize our neural network model according to Equation 1. If it is a not good fit to the data, the model should have a large estimation error before treatment. The users should also use their domain knowledge to check if the model holds for their data, i.e., if there is any factor starting to affect the outcomes in halfway and causes the representation to change over time.

## A.2  Discussion and comparison of the causal assumptions

### A.2.1  The DAG structure

Although the conventional assumption has been that the covariate $X$ directly influences the outcome and the treatment, some recent works question the plausibility of this assumption [36, 22, 55]. In many real world problems, the (high-dimensional) covariates are only a crude reflection of the underlying (low-dimensional) latent states. For instance, the covariates may contain a full array of biomarkers and medical test results, but the patient's actual health status can be represented by a handful of unobservable physiological variables.

Hence, the DAG in Figure 2 explicitly involves a latent factor $\mathbf{c}_i$. We use $\mathbf{c}_i$ to represent a patient's latent health status, which modulates both covariates and outcomes. In practice, the clinician assigns the treatment based on the observed covariates $\mathbf{x}$ (e.g. biomarkers) so we allow $\mathbf{x}$ to directly affect the treatment a.

Also note that the covariates observed *after* treatment assignment cannot causally affect the assignment. Hence, we do not adjust for any covariates observed after treatment assignment.

### A.2.2  The point treatment setting

The point treatment setting is applicable in two important scenarios. First, the drug is for treating a chronic disease so the patients tend to take the drug for a period of time after treatment initiation. Secondly, the treatment is one-off but has a long-lasting impact (stent implant or organ transplant).

Table 3: Comparison of the causal assumptions in the related works. The definitions of Consistency, Sequential overlap, and No unobserved confounder are given in A.2. The data generating process (DGP) in Equation 1 contains the one in Equation 17 as a special case.

| Approach | Ref | SUTVA | DGP | Sequential Overlap | No unobserved conf. |
|---|---|---|---|---|---|
| SC | [1] | Yes | Equation 17 | - | Yes |
| RSC | [7] | Yes | Equation 17 | - | Yes |
| MC-NNM | [8] | Yes | Equation 17 | - | Yes |
| CRN | [11] | Yes | - | Yes | Yes |
| RMSN | [35] | Yes | - | Yes | Yes |
| SyncTwin | This work | Yes | Equation 1 | - | Yes |

Importantly, the point treatment setting is still applicable when the patient receive other medical treatments over time (as long as we are not interested in estimating the causal effect of these additional medical treatments). In particular, the treatment history prior to $t = 0$ can be incorporated into the covariates $X$. The additional treatments administered after $t = 0$ cannot confound the assignment of the primary treatment at $t = 0$ and do not need to be adjusted for.

### A.2.3 Models for dynamic treatment settings

As shown in Table 3, CRN [11] and RMSN [35] makes the following three causal assumptions. (1) **Consistency**: $y_{it}(a_{it}) = y_{it}$. (2) **Sequential overlap** (aka. positivity): $Pr(a_{it} = 1|a_{i,t-1}, \mathbf{x}_{it}) > 0$ whenever $Pr(a_{i,t-1}, \mathbf{x}_{it}) \neq 0$. (3) **No unobserved confounders**: $y_{it}(0), y_{it}(1) \perp\!\!\!\perp a_{it} \mid \mathbf{x}_{it}, a_{i,t-1}$. In summary, CRN makes the same consistency (SUTVA) assumption as SyncTwin. CRN and SyncTwin both require no unobserved confounders though the assumptions take different forms. However, SyncTwin does not assume sequential overlap (or any overlap) while CRN does not make assumptions on the data generating model.

**Why does SyncTwin not explicitly require overlap**? The overlap assumption is commonly made in treatment effect estimation methods. We first give a very brief review of why two importance classes of methods need overlap. (1) For methods that rely on propensity scores, overlap makes sure that the propensity scores are not zero [35]. It thus enables various forms of propensity weighting. (2) For methods that rely on covariate adjustment [43, 11], overlap ensures that the conditional expectation $\mathbb{E}[\mathbf{y}_i|\mathbf{X}_i, a_i]$ is well-defined, i.e. the conditioning variables $(\mathbf{X}_i, a_i)$ have non-zero probability.

In comparison, SyncTwin relies on neither the propensity scores nor the explicit adjustment of covariates, and hence it does not make overlap assumption *explicitly*. However, as discussed in Proposition 2, SyncTwin requires the synthetic twin to match the representations $\mathbf{d}_i^c \approx 0$, which implies $\tilde{\mathbf{c}}_i \approx \sum_{j \in \mathcal{I}_0} b_{ij} \tilde{\mathbf{c}}_i$ for some $b_{ij}$ — the target individual should be in or close to the convex hull formed by the controls in the representation space. This condition has a similar spirit to overlap (but very different mathematically). When overlap is satisfied there tends to be control individuals in the neighbourhood of the treated individual, making it easier to construct matching twins. Conversely, if overlap is violated, the controls will tend to far away from the treated individual, making it harder to construct a good twin.

### A.2.4 Synthetic control

As shown in Table 3, Synthetic control [2, 1] and its variants [8, 7] rely on three causal assumptions: **(1)** *Stable Unit Treatment Value Assumption* [41]: $\mathbf{y}_{it}(a_i) = \mathbf{y}_{it}, \forall i \in [N], t \in \mathcal{T}^- \cup \mathcal{T}^+$. **(2)** *No anticipation*, also known as causal systems [4]: $\mathbf{y}_{it} = \mathbf{y}_{it}(1) = \mathbf{y}_{it}(0), \forall t \in \mathcal{T}^-, i \in [N]$. And **(3)** *Data generating assumption* (linear factor model):

$$\mathbf{y}_{it}(0) = \mathbf{q}_t^\top \underline{\mathbf{x}}_i + \xi_{it} \quad \forall i \in [N], t \in \mathcal{T}^- \cup \mathcal{T}^+. \tag{17}$$

where $\underline{\mathbf{x}}_i = \text{vec}(\mathbf{X}_i) \in \mathbb{R}^{D \times L}$, vec is the vectorization operation. $\xi_{it}$ is an error term that has mean zero and satisfies $\xi_{it} \perp\!\!\!\perp a_{rs}, \mathbf{x}_r$ for $\forall k, r, s, t$.

The first two assumptions are also assumed by SyncTwin.

It is worth highlighting that the data generating assumption of Synthetic Control is a **special case** of the more general assumption of SyncTwin in Equation 1. To see this, let $\mathbf{c}_i = \underline{\mathbf{x}}_i = \text{vec}(\mathbf{X}_i)$ in Equation 1, i.e. we use the flattened temporal covariates directly as the representation. Further let

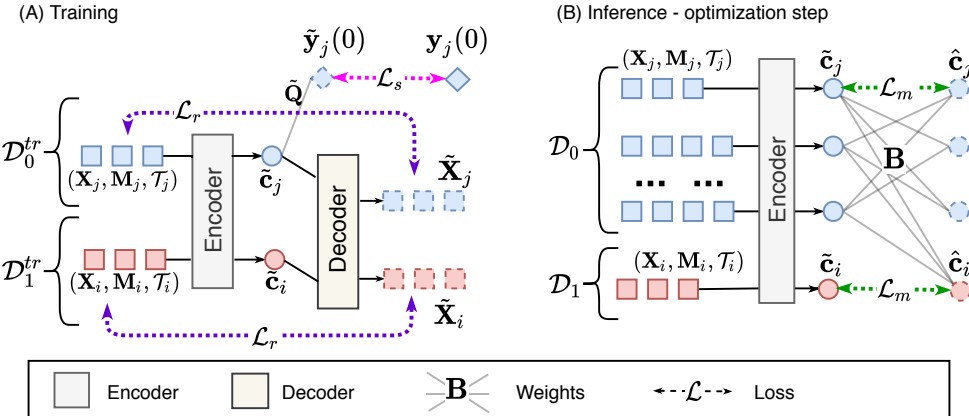

Figure 5: Illustration of the loss functions. (A) The representation networks are trained using $\mathcal{L}_s$ and $\mathcal{L}_r$ in Equation 2. Note that the supervised loss $\mathcal{L}_s$ only applies to the control. (B) Validation and inference involve optimizing the matching loss $\mathcal{L}_m$ in Equation 4. Note the encoder needs to be fixed during optimization.

$\phi_\theta(\mathbf{c}_i, t_{is}) = \mathbf{c}_i[Ds : D(s+1)]$ and $\varepsilon_{is} = 0$, where $\mathbf{c}[a:b]$ takes a slice of vector $\mathbf{c}$ between index $a$ and $b$. The result is exactly Equation 17.

### A.3  Comparison of the temporal covariates allowed in the related works

As introduced in Section 2, SyncTwin is able to handle temporal covariates sampled at different frequencies, i.e. the set of observation times $\mathcal{T}_i$ and a mask $\mathbf{m}_{it}$ can be different for different individuals. In comparison, **Synthetic Control** [2], **robust Synthetic Control** [7], and **MC-NNM** [8] are only able to handle regularly-sampled covariates, i.e. $\mathcal{T}_i = \{-1, -2, \ldots, -L\} \; \forall i \in [N]$, and $\mathbf{m}_{it} = \mathbf{1} \; \forall i \in [N], t \in \mathcal{T}_i$. In other words, the temporal covariates $[\mathbf{x}_{is}]_{s \in [S_i]} = \mathbf{X}_i \in \mathbb{R}^{D \times S_i}$ has a matrix form.

The deep learning methods including **CRN** [11] and **RMSN** [35] have the potential to handle irregularly-measured variable-length covariates when a suitable architecture is used. However, the architectures proposed in the original papers only apply to regularly-sampled case and no simulation or real data experiments were conducted for the more general irregular cases.

### A.4  Unrelated works with similar terminology

Several recent works in the deep learning ITE literature employ similar terminologies such as "matching" [29, 31]. However, they are fundamentally different from SyncTwin because they only work for static covariates and they try to match the overall distribution of the treated and control group rather than constructing a synthetic twin that matches one particular treated individual.

The Virtual Twin method [18] is designed for randomized controlled trials where there is no confounding (temporal or static). As a result, it cannot overcome the confounding bias when the problem is to estimate causal treatment effect from *observational data*.

### A.5  The generality of the assumed data generating model

SyncTwin assumes that the outcomes are generated by a *latent* factor model [48] with the *latent* factors $\mathbf{c}_i$ learnable from covariates $\mathbf{X}_i$. We assume the dimensionality of $\mathbf{c}_i$ to be low compared with the number of time steps. Despite its seemingly simple form, the assumed latent factor model is very flexible because the factors are in fact *latent factors*.

The latent factor model is widely studied in Econometrics. In many real applications, the temporally observed variables naturally have a low-rank structure, thus can be described as a latent factor model [3, 2]. The latent factor model also captures many of well-studied scenarios as special cases [17] such as the conventional additive unit and time fixed effects ($y_{it}(0) = q_t + c_i$). Last but not least, It

---

**Algorithm 1** SyncTwin training procedure.

---

**Input:** $\mathcal{D}_0^{tr}$, $\mathcal{D}_1^{tr}$
**Input:** Hyperparameters: $\lambda_r$, $\lambda_p$
**Input:** Encoder, Decoder, $\tilde{\mathbf{Q}}$
**Input:** Training iteration $max\_itr$, batch size $batch\_size$, Optimizer
Randomly initialize Encoder $\theta_e$ and Decoder $\theta_d$; set $\tilde{\mathbf{Q}} = \mathbf{0}$
**for** $itr \in (0, max\_itr]$ **do**
    Randomly draw a mini-batch of control units $\mathcal{D}_0 \subset \mathcal{D}_0^{tr}$ with $batch\_size$ samples.
    Randomly draw a mini-batch of treated units $\mathcal{D}_1 \subset \mathcal{D}_1^{tr}$ with $batch\_size$ samples.
    Evaluate training loss $\mathcal{L}^{tr}(\mathcal{D}_0, \mathcal{D}_1) = \lambda_r \mathcal{L}_r(\mathcal{D}_0, \mathcal{D}_1) + \lambda_p \mathcal{L}_s(\mathcal{D}_0)$ (defined in Equation 2).
    Calculate the gradient of $\mathcal{L}^{tr}(\mathcal{D}_0, \mathcal{D}_1)$ via back propagation.
    Update all $\theta_e$, $\theta_d$ and $\tilde{\mathbf{Q}}$ using the Optimizer.
**end for**
**Output:** Trained Encoder, Decoder and $\tilde{\mathbf{Q}}$

---

has also been shown that the low-rank latent factor models can well approximate many nonlinear latent variable models [49].

Latent factor models in the static setting are very familiar in the deep learning literature. Consider a deep feed-forward neural network that uses a linear output layer to predict some real-valued outcomes $\mathbf{y} \in \mathbb{R}^D$ in the static setting (notations used in this example are not related to the ones used in the rest of the paper). Denote the last layer of the neural network as $\mathbf{h}_{-1} \in \mathbb{R}^K$; it is easy to see that the neural network corresponds to a latent factor model i.e. $\mathbf{y} = \mathbf{A}\mathbf{h}_{-1} + \mathbf{b}$, where $\mathbf{h}_{-1}$ is the latent factor. Note that this holds true for arbitrarily complicated feed-forward networks as long as the output layer is linear.

### A.6    Estimating ITE for control individuals and applications in decision support

We have been focusing on predicting the counterfactual outcomes for a treated individual $i \in \mathcal{I}_1$. The same approach can be applied to a control individual without loss of generality. After obtaining the representation $\tilde{\mathbf{c}}_i$ for $i \in \mathcal{I}_0$, SyncTwin can use the treatment group $j \in \mathcal{I}_1$ to construct the synthetic twin by optimizing the matching loss Equation 4. The checking and estimation procedure remains the same.

To use the method in decision support, one needs to extract two corpuses of historical patients from EHR, one for the control and one for the treated. Since EHR is voluminous, we assume that both corpuses contain enough patients for constructing the twins. For a new patient $i \notin [N]$ with observed pre-treatment covariates $\mathbf{x}$ and outcomes $\mathbf{y}^-$, we can proceed as usual to estimate $\mathbf{y}(0)$ and $\mathbf{y}(1)$ from each corpus and make recommendations.

SyncTwin also easily generalizes to the situation where there are $A > 1$ treatment groups each receiving a different treatment. In this case, the treatment indicator $\mathrm{a}_i \in [0, 1, \dots, A]$. For a target individual in *any* of the treatment groups, SyncTwin can construct its twin using the control group $\mathcal{I}_0$. The remaining steps are the same as the single treatment group case.

### A.7    Detailed training, validation and inference procedure

As is standard in machine learning, we perform model training, validation and inference (testing) on three disjoint datasets, $\mathcal{D}^{tr}$, $\mathcal{D}^{va}$ and $\mathcal{D}^{te}$. We use $\mathcal{D}_0^{tr}$ and $\mathcal{D}_1^{tr}$ to denote the control and the treated in the training data and use similar notations for validation and testing data. The schematics of the architecture and various loss functions are visualized in 5.

**Training**. On the training dataset $\mathcal{D}_0^{tr}$, we learn the representation networks by optimizing $\mathcal{L}^{tr} = \lambda_r \mathcal{L}_r + \lambda_p \mathcal{L}_s$, where $\mathcal{L}_r$ and $\mathcal{L}_s$ are the loss functions defined in Equation 2. The hyperparameter $\lambda_r$ and $\lambda_p$ controls the relative importance between the two losses. We provide an ablation study in Section 5.1 and perform detailed analysis on hyperparameter importance in Appendix A.13. The objective $\mathcal{L}^{tr}$ can be optimized using stochastic gradient descent. In particular, we used the ADAM algorithm with learning rate 0.001 [33].

**Algorithm 2** SyncTwin inference procedure.

---

**Input:** Testing data set $\mathcal{D}_0^{te}, \mathcal{D}_1^{te}$
**Input:** Trained Encoder
**Input:** Training iteration $max\_itr$, batch size $batch\_size$, Optimizer
Initialize a size $|\mathcal{D}_1^{te}|$ by $|\mathcal{D}_0^{te}|$ matrix $\mathbf{B} = \mathbf{0}$ as the weight matrix.
Use Encoder to get representation $\tilde{\mathbf{c}}_i, \forall i \in \mathcal{D}_0^{te} \cup \mathcal{D}_1^{te}$
**for** $itr \in (0, max\_itr]$ **do**
    Randomly draw a mini-batch of treated units $\mathcal{D}_1 \subset \mathcal{D}_1^{tr}$ with $batch\_size$ samples.
    Evaluate matching loss $\mathcal{L}_m(\mathcal{D}_0^{te}, \mathcal{D}_1)$ (defined in Equation 4)
    Calculate the gradient of $\mathcal{L}_m(\mathcal{D}_0^{te}, \mathcal{D}_1)$ via back propagation.
    Update $\mathbf{B}$ using the Optimizer while keeping the Encoder fixed.
**end for**
Use weight matrix $\mathbf{B}$ to obtain $\hat{\tau}_i, \forall i \in \mathcal{D}_1^{te}$ using Equation 6.
**Output:** Estimated ITE $\hat{\tau}_i, \forall i \in \mathcal{D}_1^{te}$

---

**Validation**. Since we never observe the true ITE, we cannot evaluate the error of ITE estimation, $||\tau_i - \hat{\tau}_i||_2^2$. As a standard practice [11], we rely on the *factual* loss on observed outcomes: $\mathcal{L}^{va} = \sum_{j \in \mathcal{D}_0^{va}} ||\mathbf{y}_i(0) - \hat{\mathbf{y}}_i(0)||_2^2$, where $\hat{\mathbf{y}}_i(0)$ is defined as in Equation 6 and obtained as follows. We obtain the $\tilde{\mathbf{c}}_i$ for all $i \in \mathcal{D}^{va}$ and then optimize the matching loss $\mathcal{L}_m(\mathcal{D}_0^{va}, \mathcal{D}_1^{va})$ to find weights $\mathbf{b}_i^{va}$. It is important to keep the encoder fixed throughout the optimization; otherwise it might overfit to $\mathcal{D}^{va}$. Finally, $\hat{\mathbf{y}}_i(0) = \sum_{j \in \mathcal{D}_0^{va}} b_{ij}^{va} \mathbf{y}_j(0)$.

**Inference**. The first steps of the inference procedure are the same as validation. We start by obtaining the representation $\tilde{\mathbf{c}}_i$ for all $i \in \mathcal{D}^{te}$ and then obtain weights $\mathbf{b}_i^{te}$ by optimizing the matching loss $\mathcal{L}_m(\mathcal{D}_0^{te}, \mathcal{D}_1^{te})$ while keeping the encoder fixed. Using weights $\mathbf{b}_i^{te}$, the ITE for any $i \in \mathcal{D}_1^{te}$ can be estimated as $\hat{\tau}_i = \mathbf{y}_i(1) - \sum_{j \in \mathcal{D}_0^{te}} b_{ij}^{te} \mathbf{y}_j(0)$ according to Equation 6. Similarly, we obtain $\hat{\mathbf{c}}_i, \hat{\mathbf{y}}_{it}(0)$ according to in Equation 6. The expert can check $\mathbf{d}_i^y$ to evaluate the trustworthiness of $\hat{\tau}_i$.

Table 4: Parameters for each component of the architecture and the loss function for training each parameter.

| Component | Parameters | Loss function | Reference |
|---|---|---|---|
| **Attentive Encoder** | $\theta_e$ | $\mathcal{L}_s, \mathcal{L}_r$ | Section 3.2 |
| **Decoder** | $\theta_d$ | $\mathcal{L}_r$ | Section 3.2 |
| **Linear outcome prediction** | $\tilde{\mathbf{Q}}$ | $\mathcal{L}_s$ | Section 3.2 |
| **Weights** | $\mathbf{B}$ | $\mathcal{L}_m$ | Section 3.3 |

## A.8 Optimizing the matching loss

Here we present a way to optimize the matching loss $\mathcal{L}_m$ in Equation 4. To ensure the three constraints discussed in Section 3.3 while also allowing gradient-based learning algorithm, we reparameterize $\mathbf{b}_i = \text{Gumbel-Softmax}(f_m(\mathbf{z}_i), \tau)$, where $\mathbf{z}_i \in \mathbb{R}^{N_0}$, $f_m(\cdot)$ is a masking function that sets the element $z_{ii} = -\text{Inf}$ to satisfy constraint (3). $\text{Gumbel-Softmax}(\cdot, \tau)$ is the Gumbel softmax function with temperature hyper-parameter $\tau$ [28]. It is straightforward to verify that $\mathbf{b}_k$ satisfies the three constraints while the loss $\mathcal{L}_m$ remains differentiable with respect to $\mathbf{z}_k$. We use the Gumbel softmax function instead of the standard softmax function because Gumbel softmax tend to produce sparse vector $\mathbf{b}_k$, which is highly desirable as we discussed in Section 3.

The memory footprint to directly optimize $\mathcal{L}_m$ is $O\big((|\mathcal{D}_0| + |\mathcal{D}_1|) \times |\mathcal{D}_0|\big)$, which can be further reduced to $O\big(|\mathcal{D}_B| \times |\mathcal{D}_0|\big)$ if we use stochastic gradient decent with a mini-batch $\mathcal{D}_B \subseteq \mathcal{D}_0 \cup \mathcal{D}_1$.

## A.9 Implementation details of the benchmark algorithms

**Synthetic control**. We used the implementation of Synthetic Control in the `R` package `Synth` (1.1-5). The package is available at `https://CRAN.R-project.org/package=Synth`.

**Robust Synthetic Control**. We used the implementation accompanied with the original paper [7] at `https://github.com/SucreRouge/synth_control`. We optimized the hyperparame-

ters on the validation set using the method described in Section 3.4.3 [7]. The best hyperparameter setting was then applied to the test set.

**MC-NNM**. We used the implementation in the `R` package `SoftImpute` (1.4) available at `https://CRAN.R-project.org/package=softImpute`. The regularization strength $\lambda$ is tuned on validation set using grid search before applied to the testing data.

**Counterfactual Recurrent Network** and **Recurrent Marginal Structural Network**. We used the implementations by the authors [11, 35] at `https://bitbucket.org/mvdschaar/mlforhealthlabpub/src/master/`. The networks were trained on the training dataset. We experimented different hyper-parameter settings on the validation dataset, and applied the best setting to the testing data. We also found that the results are not sensitive to the hyperparameters.

**Counterfactual Gaussian Process**. We used the implementation with GPy [19], which is able to automatically optimize the hyperparameters such as the kernel width using the validation data.

**One-nearest neighbour**. We used our own implementation. Since no parameters need to be learned or tuned, the algorithm was directly applied on the testing dataset.

Search range of hyper-parameters

1. Synthetic control: hyperparameters are optimized by `Synth` directly.
2. Robust Synthetic control: num_sc $\in \{1, 2, 3, 4, 5\}$
3. MC-NNM: $C \in \{3, 4, 5, 8, 10\}$
4. Counterfactual Recurrent Network: max_alpha $\in \{0.1, 0.5, 0.8, 1\}$, hidden dimension $H \in \{32, 64, 128\}$
5. Recurrent Marginal Structural Network: hidden dimension $H \in \{32, 64, 128\}$
6. Counterfactual Gaussian Process: hyperparameters are optimized by `GPy` directly.

## A.10    The simulation model

In Equation 9, $R_t$ is the LDL cholesterol level (outcome) and $I_t$ is the dosage of statins. For each individual in the treatment group, one dose of statins (10 mg) is administered daily after the treatment starts, which gives dosage $I_t = 0$ if $t \leq t_0$ and $I_t = 1$ otherwise. $K$, $H$ and $D_{50}$ are constants fixed to the values reported in [16]. $K_t^{in} \in \mathbb{R}$ is a individual-specific time varying variable that summarizes a individual's physiological status including serum creatinine, uric acid, serum creatine phosphokinase (CPK), and glycaemia. $P_t$ and $D_t$ are two intermediate temporal variables both affecting $R_t$.

## A.11    Additional simulation results

Table 5: Mean absolute error on ITE with varying irregular $m$. $S = 25$ and $p_0 = 0.5$ are used in all cases. Estimated standard deviations are shown in the parentheses. The best performer is in bold. * did not finish within 48h.

| Method | $N_0 = 200$ | | | $N_0 = 1000$ | | |
|---|---|---|---|---|---|---|
| | $m = 0.7$ | $m = 0.5$ | $m = 0.3$ | $m = 0.7$ | $m = 0.5$ | $m = 0.3$ |
| SyncTwin-Full | **0.129** (.009) | **0.143** (.010) | **0.190** (.012) | **0.110** (.006) | **0.116** (.006) | **0.141** (.008) |
| SyncTwin-$\mathcal{L}_r$ | 0.158 (.013) | 0.177 (.014) | 0.245 (.017) | 0.126 (.007) | 0.133 (.008) | 0.175 (.012) |
| SyncTwin-$\mathcal{L}_s$ | **0.128 (.010)** | 0.157 (.011) | 0.234 (.016) | 0.140 (.009) | 0.135 (.009) | 0.174 (.013) |
| SC | 0.155 (.017) | 0.201 (.016) | 0.327 (.023) | 0.145 (.015) | 0.215 (.020) | 0.359 (.026) |
| RSC | 0.415 (.021) | 0.521 (.028) | 0.640 (.044) | * | 0.495 (.028) | * |
| MC-NNM | 0.363 (.020) | 0.556 (.031) | 0.898 (.050) | 0.174 (.010) | 0.332 (.021) | 0.556 (.036) |
| CFRNet | 0.321 (.030) | 0.303 (.018) | 0.484 (.032) | 0.139 (.009) | 0.202 (.013) | 0.267 (.018) |
| CRN | 0.274 (.020) | 0.318 (.026) | 0.512 (.032) | 0.663 (.044) | 0.448 (.029) | 0.542 (.040) |
| RMSN | 0.350 (.028) | 0.370 (.025) | 0.418 (.030) | 0.401 (.032) | 0.426 (.033) | 0.479 (.035) |
| CGP | 0.568 (.037) | 0.553 (.037) | 0.631 (.045) | 0.605 (.039) | 0.626 (.039) | 0.689 (.044) |
| 1NN | 1.584 (.080) | 1.725 (.098) | 1.703 (.096) | 1.455 (.084) | 1.680 (.088) | 1.531 (.089) |

Table 5 shows the results under irregularly-measured covariates with varying degree of irregularity $m$ (smaller $m$, more irregular and fewer covariates are observed). For methods that are unable to

Table 6: Mean absolute error on ITE under different lengths of the temporal covariates $S$. $m = 1$ and $p_0 = 0.5$ are used in all cases. Estimated standard deviations are shown in the parentheses. The best performer is in bold. * did not finish within 48 hours.

| Method | $N_0 = 200$ | | | $N_0 = 1000$ | | |
|---|---|---|---|---|---|---|
| | $S = 15$ | $S = 25$ | $S = 45$ | $S = 15$ | $S = 25$ | $S = 45$ |
| SyncTwin-Full | **0.121** (.009) | **0.128** (.008) | **0.120** (.007) | **0.097** (.005) | **0.094 (.005)** | **0.085** (.004) |
| SyncTwin-$\mathcal{L}_r$ | 0.170 (.014) | 0.135 (.010) | 0.139 (.010) | 0.114 (.006) | 0.102 (.006) | 0.098 (.006) |
| OURS-$\mathcal{L}_s$ | 0.130 (.010) | **0.119 (.008)** | 0.123 (.008) | 0.119 (.007) | 0.127 (.010) | 0.106 (.007) |
| SC | 0.140 (.019) | 0.149 (.018) | 0.138 (.021) | 0.190 (.029) | 0.214 (.036) | 0.215 (.044) |
| RSC | 0.348 (.023) | 0.322 (.019) | 0.228 (.011) | * | 0.302 (.014) | * |
| MC-NNM | 0.454 (.023) | 0.226 (.011) | 0.159 (.008) | 0.140 (.007) | 0.124 (.006) | 0.109 (.005) |
| CFRNet | 0.316 (.025) | 0.291 (.003) | 0.143 (.008) | 0.353 (.035) | 0.104 (.007) | 0.095 (.005) |
| CRN | 0.307 (.022) | 0.335 (.023) | 0.316 (.022) | 0.282 (.018) | 0.563 (.035) | 0.457 (.028) |
| RMSN | 0.311 (.028) | 0.334 (.027) | 0.493 (.032) | 0.342 (.032) | 0.390 (.032) | 0.557 (.036) |
| CGP | 0.561 (.036) | 0.561 (.035) | 0.549 (.035) | 0.578 (.037) | 0.602 (.038) | 0.611 (.038) |
| 1NN | 1.356 (.072) | 1.614 (.078) | 1.575 (.078) | 1.322 (.072) | 1.384 (.083) | 1.744 (.098) |

Table 7: Sparsity metrics of the learned $\mathbf{b}_i$. Estimated standard deviations are shown in the parentheses. Here $p_0 = 0.5, m = 1, S = 25$. The worst performer is italicized

| Method | $N_0 = 200$ | | | $N_0 = 1000$ | | |
|---|---|---|---|---|---|---|
| | Gini | Entropy | N Control | Gini | Entropy | N Matched |
| SyncTwin-Full | .207 (.017) | .394 (.032) | 1.745 (.070) | .242 (.017) | .482 (.034) | 1.830 (.073) |
| SyncTwin-$\mathcal{L}_r$ | .196 (.016) | .381 (.031) | 1.710 (.070) | .267 (.018) | .548 (.037) | 1.930 (.080) |
| SyncTwin-$\mathcal{L}_s$ | .213 (.016) | .409 (.030) | 1.760 (.068) | .306 (.018) | .631 (.039) | 2.080 (.086) |
| SC | *.792 (.009)* | *1.871 (.035)* | *6.125 (.135)* | *.862 (.006)* | *2.274 (.029)* | *7.059 (.110)* |

deal with irregular covariates, we first impute the unobserved values using Probabilistic PCA before applying the algorithms [23]. SyncTwin achieves the best performance in all cases. Furthermore, SyncTwin's performance deteriorates more slowly than the benchmarks when sampling becomes more irregular (larger $m$). This suggests that the encoder network in SyncTwin is able to learn good representations even from highly irregularly-measured sequences. Table 6 shows the results under various lengths of the observed covariates $S$ (smaller $S$, shorter sequences are observed). Again SyncTwin achieves the best performance in all cases. As expected, SyncTwin makes smaller error when the observed sequence is longer. Note that this is not the case of CRN and RMSN — their performance deteriorates when the observed sequence is longer. This might indicate that these two methods are less able to learn good balancing representations (or balancing weights) when the sequence is longer.

## A.12 Sparsity compared with Synthetic Control

In Figure 3 (A) we have shown visualy that SyncTwin produces sparser solution than SC. To quantify the differences, we report the Gini index ($\sum_{ij} \mathbf{b}_{ij}(1 - \mathbf{b}_{ij})/N_1$), entropy ($\sum_{ij} -\mathbf{b}_{ij}\log(\mathbf{b}_{ij})/N_1$) and the number of contributors used to construct the twin ($\sum_{ij} \mathbf{1}\{\mathbf{b}_{ij} > 0\}/N_1$) in the simulation study. All three metrics reflect the sparsity of the learned weight vector (smaller more sparse). Table 7 shows that SyncTwin achieve sparser results that SC in all metrics considered. The full and ablated versions of SyncTwin have similar sparsity because the sparsity is regulated in the matching loss, which all versions share. It is worth pointing out that RSC and MC-NNM do not produce sparse weights and the weights do not need to be positive and sum to one [7, 8].

## A.13 Sensitivity of Hyper-Parameters

It is beneficial to understand the network's sensitivity to each hyper-parameter so as to effectively optimize them during validation. In addition to the standard hyper-parameters in deep learning (e.g. learning rate, batch size, etc.), SyncTwin also includes the following specific hyper-parameters: (1) $\tau$, the temperature of the Gumbel-softmax function Appendix A.8, (2) $\lambda_p$ in the training loss $\mathcal{L}^{tr}$ (since

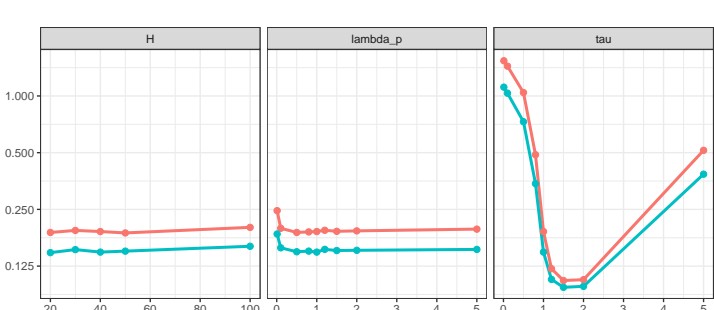

Figure 6: The sensitivity of hyper-parameters on the mean absolute error of ITE estimation and the validation loss defined in Section 3.5. The left panel shows the results for various choices of $H$; the middle panel shows the ratio between $\lambda_p$ and $\lambda_r$; and the right panel shows $\tau$. The y-axis is shown in log scale.

only the ratio between $\lambda_p$ and $\lambda_r$ matters, we keep $\lambda_r = 1$ and search different values of $\lambda_p$) , and (3) $H$, the dimension of the representation $\tilde{\mathbf{c}}_i$.

Here we present a sensitivity analysis on the hyper-parameters $H$, $\lambda_p$ and $\tau$ using the simulation framework detailed in Section 5.1. Here we present the results for $N_0 = 2000$ and $S = 15$ although these results generalize to all the simulation settings we considered. The results are presented in Figure 6, where we can derive two insights.

Firstly, the hyper-parameter $\tau$ is very important to the performance and need to be tuned carefully during validation. This is understandable because $\tau$ is the temperature parameter of the Gumbel softmax function and it directly controls the sparsity of matrix $\mathbf{B}$. In comparison, hyper-parameter $H$ and $\lambda_p$ do not impact the performance in significant way. Therefore we recommend to use $H = 40$ and $\lambda_p = 1$ as the default.

Secondly, we observe that the validation loss $\mathcal{L}^{va}$ closely tracks the error on ITE estimation (which is not directly observable in reality). These results support the use of $\mathcal{L}^{va}$ to validate models and perform hyper-parameter optimization.

## A.14 Computation time

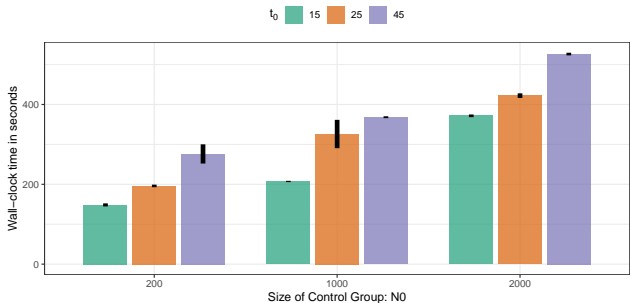

Figure 7: The wall-clock time of the simulation study under different settings. For each setting, 10 independent simulation runs were conducted. The bar shows the average wall-clock time and the line range captures the 95% confidence interval.

In figure 7 we present the wall-clock computation time (in seconds) of SyncTwin under various simulation conditions — with the control group size $N_0 = (200, 1000, 2000)$ and the length of pre-treatment period $S = (15, 25, 45)$. The simulations were performed on a server with a Intel(R) Core(TM) i5-8600K CPU @ 3.60GHz and a Nvidia(R) GeForce(TM) RTX 2080 Ti GPU. All simulations with SyncTwin finished within 30 mins. As we expect, the computation time increases with respect to $N_0$ and $S$ as more data need to be processed. However, a 10-fold increase in $N_0$ only

approximately doubled the computation time, suggesting that SyncTwin scales well with sample size. In comparison, $S$ seems to affect the computation time more because the encoder and decoder need to be trained on longer sequences.

## A.15 Description and summary statistics of the CPRD data

Access to the CPRD is regulated. We have signed an end user license before access to the data was granted. All patient records were pseudonymized in CPRD.

Any treatment effect estimation algorithm could be used negatively if the user intentionally chooses to worsen the outcome. This is very unlikely in our case because the intended users of SyncTwin are clinicians and medical researchers.

The treatment and the control group in the CPRD experiment are selected based on the selection criterion in Figure 8. We have followed all the guidelines listed in [13] to make sure the selection process does not increase the confounding bias. The summary statistics of the treatment and control groups are listed below. We can clearly see a selection bias as the treatment group contains a much higher proportion of male and people with previous cardiovascular or renal diseases.

Table 8: The summary statistics of the treatment and control groups

|  | Treatment Group | Control Group |
|---|---|---|
| % male | 59% | 51% |
| Median age | 61 | 60 |
| Townsend Index | 8 | 8 |
| % CVD | 16% | 9% |
| % Renal disease | 16% | 12% |
| % Atrial Fibrillation | 4% | 4% |

## A.16 Cohort selection criterion in the CPRD study

Refer to the figure at the end (page 26).

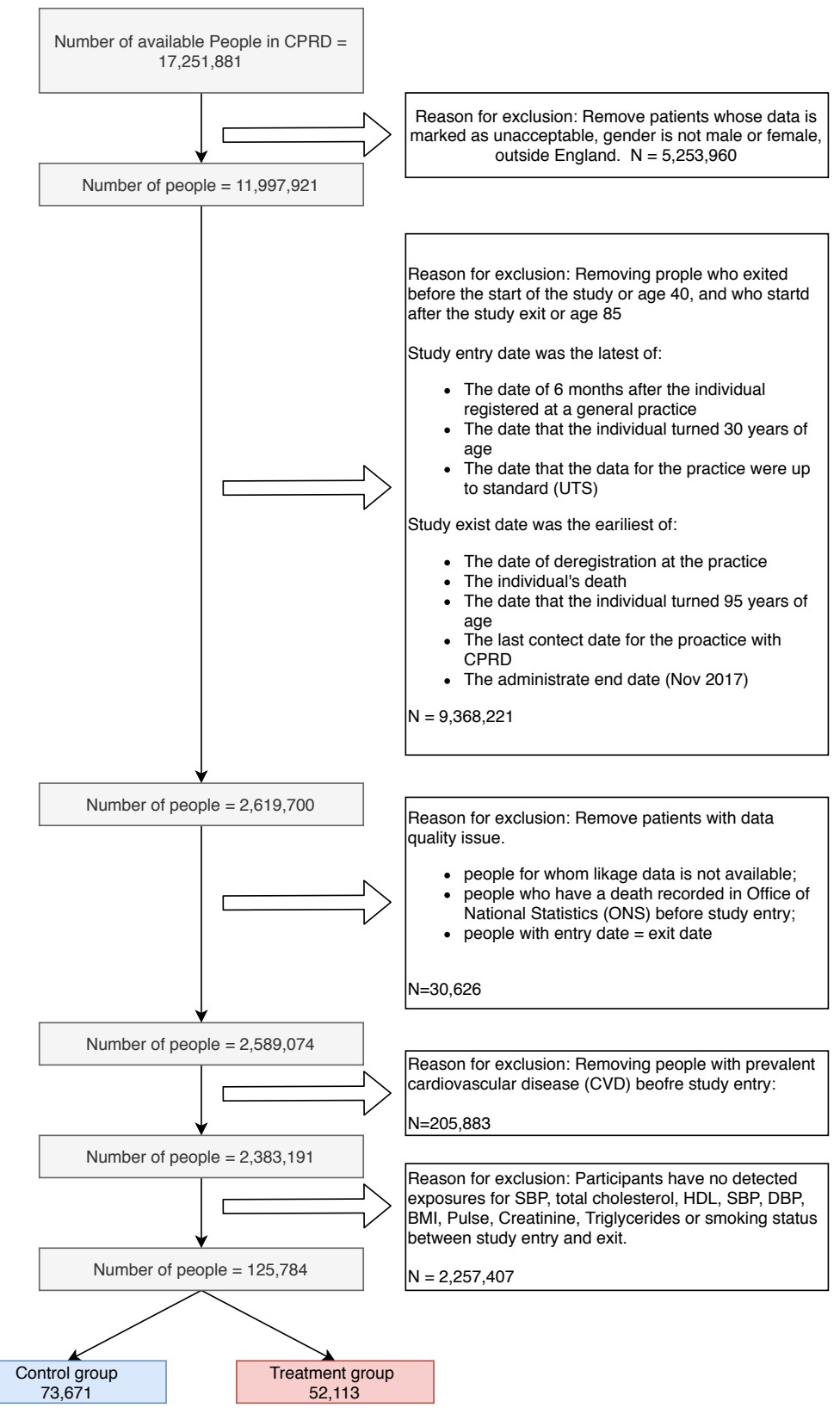

Figure 8: Flowchart for selection of eligible individuals from CPRD for the observational study on the treatment effect of statins. Numbers represent unique individuals in each group.