# OpenReview forum: "SyncTwin: Treatment Effect Estimation with Longitudinal Outcomes"
_NeurIPS.cc/2021/Conference — NeurIPS 2021 Poster_

### Official Review · Reviewer_tEce · 2021-07-14

**Rating:** 8
**Confidence:** 3

**Summary:**

This paper proposes a method, SyncTwin, to estimate treatment effects in Longitudinal and Irregularly sampled data with Point treatment (LIP) setting. The idea is to create a synthetic twin from controls that closely matches a treated target in representation. By using the pretreatment time samples of the twin to estimate temporal trend, SyncTwin can predict the outcome of the target when no treatment is administered. The method is applied to simulated and real data with lower prediction error shown.


**Ethical Concerns:**

The paper does not seem to have ethical issues.

**Limitations And Societal Impact:**

The authors mentioned extension to dynamic treatment settings and model outcomes in continuous time. The authors also mentioned, in practice, the clinician can accept the recommended treatment when the error bound is below a threshold and resort to expert knowledge otherwise. The question is how reliable/robust is the error bound to outlier subjects/measurements.

**Main Review:**

Originality
Not sure if the combined idea of creating a synthetic twin and using pretreatment outcome is novel since Synthetic Control at a high level seems to be doing that as well.

Quality
The method seems sound, and the results on both simulated and real data are solid. A couple of question comes to mind. First, are there studies where no matched control data are collected during post-treatment period (which is used as ground truth in the current paper)? Second is what happens if we directly use the outcome prediction without creating a twin, since SyncTwin provides such output?

Clarity
The paper is mostly clear with good motivation provided for the proposed method, i.e. better use of pretreatment outcomes. Also, dividing the paper into sections on problem statement, assumptions, etc. make the paper easy to read. The problem statement is clear, but would suggest labeling Figure 1 with variable names. Also, the sentence “the pre-treatment outcomes are arguably much more closely linked to the outcomes after treatment than the covariates— they hence deserve special considerations, e.g. modifying the architecture or loss function to reflect their importance” is a bit unclear without proper context in the Introduction.

Significance
Being able to predict treatment outcome is of broad interest. The method provides both outcome prediction and error estimates to inform clinicians. For others to adopt this method, the authors will need to describe in more details on what kind of data are applicable, i.e. this method would not be needed if we have data from matched controls taking placebo.

Summary Post Author Response
I have read through the reviews and responses. I will keep my original score, which was mainly based on the results. My main concern was real world applications. It might be worth adding a bit more details on how to select a matched control group for executing the proposed method when data are not from well designed experiments, i.e. a random set of controls might not match well with all patients.

**Time Spent Reviewing:**

6

---

> ### Author Response · Authors · 2021-08-10
> **Response to Reviewer tEce**
>
> Thank you for your thoughtful comments and suggestions. We’ve provided a point-by-point response below. Please let us know if anything needs further clarification.
>
> ---
>
>
>
> 1. SyncTwin requires the outcomes to be observed for both groups. This is a standard (and minimum) requirement -- if one never observes the outcome for any control individual, it would be impossible to estimate $Y(0)$  from data.
>
>
> 2. Direct outcome prediction (without creating matching twins) cannot adjust the distribution shift between the treated and control groups. In fact, the DR-CFR benchmark attempts to improve direct regression by adding regularization terms. Hence, we expect direct regression to under-perform the DR-CFR benchmark (DR-CFR underperforms SyncTwin in the experiments).
>
>
> 3. We will incorporate the proposed labelling and textual changes in the revision.

---

> > ### Comment · Reviewer_tEce · 2021-08-20
> > **Typical Dataset**
> >
> > For 1., the question is whether there are studies where no matched control data are collected during the *post treatment* period, i.e. I assume controls would be taking placebo and data would be collected after taking the placebo. Basically, I am trying to understand the real world scenarios at which the proposed method would be applied.

---

> > > ### Author Response · Authors · 2021-08-25
> > > **Further response to Reviewer tEce**
> > >
> > > Thanks for your reply and clarification.
> > >
> > > Placebo is typically only assigned in randomized controlled trials, but not in the routine clinical practice. The observational data collected from EHR will not contain patients with placebo. Hence, we define the control group as those who did *not* receive the treatment (rather than those who received the placebo).
> > >
> > > If placebo effect does exist, the observational study may fail to estimate the true effect. However, in most existing observational studies the placebo effect is assumed to be negligible compared with the true effect unless strong evidence points otherwise [1].
> > >
> > > [1] Rosenbaum, Paul R., and Donald B. Rubin. "Assessing sensitivity to an unobserved binary covariate in an observational study with binary outcome." Journal of the Royal Statistical Society: Series B (Methodological) 45.2 (1983): 212-218.

---

### Official Review · Reviewer_Hv4x · 2021-07-16

**Rating:** 5
**Confidence:** 3

**Summary:**

The paper proposes an individual treatment effect estimation method called SyncTwin, which learns synthetic twins of target patients via the use of representation learning for counterfactual predictions. The proposed method is used on longitudinal data such as electronic health records. The paper is concerned with the setting of longitudinal and irregularly sampled data with point treatment. The paper describes the proposed method, offers some theoretical guarantees, and demonstrates the utility of the method on both synthetic and real-world data.

**Ethical Concerns:**

I am somewhat concerned about the depolyment of the method in the real world because of the nature of individual treatment effect estimation. The reliability of such methods should be held to a high standard and guidelines and failsafe should be considered to make sure that the results of the proposed method are used responsibly in decision making.

**Limitations And Societal Impact:**

I think the authors can discuss more the limitation of the proposed method and understand the failure mode of the proposed method better.

**Main Review:**

I think the paper makes efforts to address an important problem in causal inference, i.e. individual treatment effect estimation from longitudinal data. The proposed method is concerned with a specific, relatively simple setting (compared to dynamic treatment ) without unobserved confounders.

Originality: I think the proposed method is original, in terms of deriving solutions tailored to the specific setting proposed in the paper. The proposed method is also closely related to several lines of research in causal inference but different from those in various ways.

Quality: I think the proposed method presents solutions to deal with various important challenges under the specific setting. Theoretical characterization is provided in the paper, and the analysis of experimental results on both synthetic data and real-world data is detailed.  I am somewhat unconvinced by the interpretability provided by the method. While the linear combinations of contributors provide some convenience to interpretability, such contributors are learned latent representations, which potentially are difficult to interpret. While the proposed method is validated on real-world data, the authors use the fact that the average treatment effect learned by the method is close to the one reported in RCT as an important argument to justify the effectiveness of the proposed method. I am not sure if such a justification suffices. For one thing,  it is unclear whether the population in the observational study is comparable to that in the RCT (despite the authors used the admission criteria of the RCT for the observation study).  It is also unclear whether the observational study meets the LIP setting, satisfies the assumptions made in the paper, and whether all confounders are appropriately adjusted. Based on these concerns, I am somewhat skeptical about the reliability of the results from real-world data.

Clarity: I think the paper is clear. Notation are well defined and used.

Significance: I believe the paper has certain significance based on the setting that it deals with and the difference it has compared to other related work. It's also a good attempt at tackling more challenging causal inference problems from longitudinal data.

**Time Spent Reviewing:**

2

---

> ### Author Response · Authors · 2021-08-10
> **Response to Reviewer Hv4x**
>
> Thank you for your thoughtful comments and suggestions. We’ve provided a point-by-point response below. Please let us know if anything needs further clarification.
>
> ---
>
> ### (1) Interpretability
>
> SyncTwin adopts the notion of “data point interpretability” [1], where one explains the prediction by presenting the most relevant data points (X, Y) to the users. SyncTwin issues prediction via a convex combination of the control outcomes (Equation 6). Hence, the most relevant data points are (by definition) the individuals with the highest weights $b_j$ -- we can present their covariates and outcomes to the clinicians to make interpretation. In contrast, blackbox ML-based estimators cannot directly provide this type of interpretation.
>
> Note that the learned representations $\tilde{c}$ are internal to the algorithm; we do not intend to show these representations to the clinician or to make interpretations on them.
>
> ### (2) Real-data experiment
>
> We have relied on epidemiologists with expertise in medical observational studies to design and carry out the real-data study. We did not acknowledge any contributing epidemiologists in the current version because of the anonymity requirement, but we will do so in the final version.
>
> Specifically, we have selected the patient cohort that is comparable to the Heart Protection Study trial -- we use the same admission criteria and the key baseline covariates are similar in distribution (we will expand table 8 to compare the summary statistics of the two studies in the revision). Based on the domain knowledge, we selected eight temporal covariates that are likely to confound the treatment assignment (line 308). The study design adheres to the LIP setting exactly.
>
> We stress that few existing works in the literature validates the method by conducting a real-world observational study to replicate the findings of a randomized controlled trial. Most existing works rely on synthetic or semi-synthetic data for validation, which may significantly under-represent the complexity of the real-world data. We fully agree with the reviewer that the causal inference methods need to be evaluated with a high standard -- and we believe that our real-data experiment is a step towards that goal.
>
> ---
>
> ### References
>
> [1] Stiglic, Gregor, et al. "Interpretability of machine learning‐based prediction models in healthcare." Wiley Interdisciplinary Reviews: Data Mining and Knowledge Discovery 10.5 (2020): e1379.

---

> ### Author Response · Authors · 2021-08-17
> **Dear Reviewer Hv4x**
>
> Once again, thank you for your invaluable feedback. We were wondering whether our response has addressed your concerns. If you have any additional comments, please let us know, we would be happy to address them.

---

### Official Review · Reviewer_n3BB · 2021-07-17

**Rating:** 5
**Confidence:** 3

**Summary:**

The paper proposes a representation learning approach to model counterfactuals in longitudinal data with point interventions over time. Their approach, SyncTwin is thus suited to situations where an intervention is made on a temporal dataset at an instance of time or as a continuous exposure by treating all points post the intervention as the potential outcomes under treatment.

They compare the performance of their model against several counterfactual estimation baselines originally proposed to model static observational data with interventions.

**Limitations And Societal Impact:**

I would encourage the authors to rephrase the Societal Impact section by focussing on how accurate counterfactual estimation can better help policy making for decision support in critical scenarios such as healthcare.



**Main Review:**

The problem of identifying the *Individual Treatment Effect* in temporal settings is a challenging one due to issues of identifiability of the potential outcome over time. In this paper, the authors assume a simple setting with a binary treatment or intervention, such that all points post the intervention are considered as sampled from the distribution under treatment. Furthermore, at each time step the treatment is considered the same (the current approach does not model the effect of time-varying interventions)

Their main innovation is that at test time, the counterfactual outcome is determine by modelling it as mixture model over the observed time-steps (in the paper they refer to the weights of this mixture as $b_{ij}$ ).

## Strengths

- Interesting problem of counterfactual inference with longitudinal data.
- Some theoretical insights around the latent factor model maybe interesting to the readers.
- Extensive experimental evaluation against multiple baselines on synthetic and real world datasets.

## Major Concerns / Weaknesses

- The authors frequently confuse the term, *Data Generation Process* (DGP) and the *Modeling Assumption*.(Ex. lines 135-136) The DGP that there is a latent factor $c_i$ that causes the outcome and covariates jointly, is different from the modeling assumption that the relationship is linear. The authors should reconcile this terminology and clearly differentiate between the modeling assumptions and the data generation process.

- The representation $c_i$ is frequently referred to as a *latent variable*. For $c_i$ to be a latent variable, there needs to be some explicit distributional assumption on $c_i$, which is not the case in the assumed model. The authors should thus instead call it a representation, not a latent variable.

- The authors motivate the use of their mixing weights $b_i$ from an optimization perspective. In the DAG notation however, the $b_i$ s are mixing weights or probabilities. The authors should reconcile the proposed model clearly specifying the modeling assumptions via another DAG, (ideally separate from the DGP in figure 2) that includes $b_i$ as a *latent variable* over the time-steps.

-  The final concern this reviewer has is that the current framework does not allow the model to represent the potential outcome $Y(1)$ or $Y(0)$ with a non-linear function of the representation $c_i$. The authors mention that this would be "inconsistent with the DGP" (ln 135)". Firstly, as noted above, this is a modelling assumption, not the DGP. Secondly, this reviewer believes that this assumption is not required per se under the proposed model, if the authors were to remodel $c_i$, factual outcomes $y_i$ and mixing weights $b_i$ as true random variables (with explicit probabilistic interpretation). The Matching Loss and Reconstruction Loss would thus be easily modified as a likelihood in that case.

Overall this reviewer believes that although the proposed approach is interesting, in its current form the paper requires substantial revision to:
- present and motivate the approach in a formal probabilistic sense, including the mixing weights $b_i$ and the representation $c_i$
- De-confound the DGP from the Modeling Assumptions.

## Other concerns:

An important baseline that seems to be omitted when benchmarking performance is the Causal Effect-VAE (CEVAE) (citation [35] in the paper). I would recommend the authors to explicitly compare SyncTwin against CEVAE with an LSTM based encoder-decoder (to obtain the latent $c_i$)



**Time Spent Reviewing:**

2

---

> ### Author Response · Authors · 2021-08-10
> **Response to Reviewer n3BB**
>
> Thank you for your thoughtful comments and suggestions. We’ve provided a point-by-point response below. Please let us know if anything needs further clarification.
>
> ---
>
> ### (1) Terminology: data generating process and modelling assumption
>
> We regret that the reviewer was confused about the terminology, data generating process (DGP). The term DGP was introduced in the item (3) of the assumption list in *Section 3.1 Assumptions*. Hence our assumptions about DGP is *part* of our modeling assumptions.
>
> We emphasize that it is common in the literature to make *assumptions* about DGP [1,2]. In real applications, the *true* DGP is almost always unknown or intractable, e.g. the health outcome is determined by complex physiological processes and environmental factors. Hence, one would typically assume a simplified and approximated DGP.
>
> ### (2) Terminology: variable and representation
>
> We used the term “representation” to denote the value learned by the encoder network $\tilde{c}$, as in “representation learning”. We used  the term “variable” to denote the corresponding true value ${c}$, which is unobservable.
>
> To highlight the fact that we do not make any distributional assumption on ${c}$, we will rename ${c}$ as a latent “factor” in the revision.
>
> ### (3) Mixing weights
>
> We emphasize that the mixing weights $b_i$ are *not* part of the DAG presented in Figure 2, neither is it involved in any assumptions we made in Section 3.1. The $b_i$ are deterministic weights not random variables (line 153). They are learned by optimization (Equation 4) such that the target individual’s representations are *reconstructed* via a convex combination of control individuals.
>
> We stress that using deterministic weights is a standard practice in the synthetic control literature (Appendix 3.1), to which this work conforms. Although there may exist a probabilistic interpretation of these weights, it is not the primary focus of the current work.
>
> ### (4) Nonlinearity
>
> We reiterate that, under the assumptions in Section 3.1, SycnTwin allows a *nonlinear* relationship between the covariates $X$ and the outcomes $Y$ through the latent factor $c$. Such nonlinearity is common in medical datasets and SycnTwin is designed to model that.
>
> We posit a linear relationship between $Y$ and the *unobservable* $c$ -- but do *not* assume linearity between $c$ and $X$. As we show in Appendix 5, this does not significantly reduce model flexibility. In fact, any neural network with a linear output layer assumes linearity between the outcome and the last hidden layer.
>
> The linearity between $c$ and $Y$ is fundamental to SyncTwin. Under this assumption,  the weights $b_i$, which can *linearly* reconstruct $c$, can also predict the outcome $Y$ (Equation 5 and 6). In fact, this is the basis of Proposition 3,  which provides a solution for controlling the estimation error; also see line 277 for experiments on error control.
>
> We would like to stress that linearity and prior distribution are two unrelated aspects. The linearity between $c$ and $Y$ is still necessary even when one places a prior distribution on $c$ or $b_i$. For an illustrative analogy, consider the standard PCA and the probabilistic PCA [3]. Even with a prior distribution, probabilistic PCA can only perform *linear* dimension reduction.
>
> ---
>
> ### References
>
> [1] Hansen, Bruce E. "Challenges for econometric model selection." Econometric Theory 21.1 (2005): 60-68.
>
> [2] Ullman, Jodie B., and Peter M. Bentler. "Structural equation modeling." Handbook of Psychology, Second Edition 2 (2012).
>
> [1] Bishop, Christopher M. "Bayesian PCA." Advances in neural information processing systems (1999): 382-388.

---

> ### Author Response · Authors · 2021-08-17
> **Dear Reviewer n3BB**
>
> Once again, thank you for your invaluable feedback. We were wondering whether our response has addressed your concerns. If you have any additional comments, please let us know, we would be happy to address them.

---

### Official Review · Reviewer_CDQY · 2021-07-17

**Rating:** 6
**Confidence:** 4

**Summary:**

The paper introduces a latent variable model (SyncTwin) for estimating time-varying individualized (point-) treatment effects (ITE) given temporal irregularly sampled pre-treatment covariates and outcomes. Theoretical results justifying the proposed approach include upper bound errors on the learned representations and counterfactual predictions. Experimental results on synthetic and real-world datasets show SyncTwin outperforms baselines per mean absolute error (MAE) on ITE and recovering RCT findings, respectively.

**Limitations And Societal Impact:**

The discussion on limitations needs to be expanded.

**Main Review:**

Overall I enjoyed reading this paper; the reviewer appreciates the clarity and organization.

**Strengths**
- The formulated latent variable model encodes interpretability by leveraging representation learning and a matching algorithm for counterfactual prediction.
- The theoretical analysis for upper bounds on the learned representations and counterfactual predictions justify the proposed formulation.
-  The paper shows SyncTwin recovers RCT findings on a challenging, large-scale real-world EHR dataset, something rarely done in most machine learning papers.

*Below are the limitations.*

**Technical limitations**
- The paper seems like a straightforward extension of Synthetic control (SC), albeit accounting for irregularly sampled covariates. Also, the performance gains over SC seem marginal given the complexity of the proposed approach.
- Unlike approaches such as CFRNet, SyncTwin is memory inefficient as it requires the storage of learned representations from the training datasets for the matching algorithm at test time.
- Given that the matching algorithm is key to SyncTwin, empirical/theoretical analysis of best or worse case scenarios would strengthen the paper.

**Weak experiments**
-  Experimental findings on the real-world dataset are missing baseline results from SC or I-NN?
-  Also, the paper should provide qualitative results from synthetic datasets showing predicted $y_t$ against ground truth.
- Comparisons with propensity weighted approaches would be interesting.


**Time Spent Reviewing:**

4

---

> ### Author Response · Authors · 2021-08-10
> **Response to Reviewer CDQY**
>
>
> Thank you for your thoughtful comments and suggestions. We’ve provided a point-by-point response below. Please let us know if anything needs further clarification.
>
> ---
>
> ### (1) Improvements over Synthetic Control
>
> SyncTwin makes two major improvements over Synthetic Control (SC). First, as the reviewer rightly pointed out, it is able to handle irregularly observed covariates. More importantly, SyncTwin relaxes the linear assumption of SC and allows the covariates to influence the outcomes in a *nonlinear* way through the variable $c_i$ (see the discussion in Appendix 3.1).
>
> Both aspects are crucial for successful applications to real world medical datasets where irregular sampling and nonlinearity are common.
>
> Furthermore, we have articulated the necessary assumptions for SyncTwin and developed guarantees for error control. Hence, SyncTwin is a theoretically-grounded method (rather than an ad-hoc extension of SC).
>
> ### (2) Memory efficiency
>
> We agree that SyncTwin is less memory efficient at run time than approaches like CFRNet. However, memory availability is rarely a *bottleneck*. For instance, it takes approximately *400 MB*  memory to store the representations for 1,000,000 patients (assuming 100 dimensional representations and 32 bit float point numbers). Modern computers should easily satisfy this memory requirement.
>
> ### (3) Theoretical analysis on estimation performance
>
> Proposition 3 (line 194) provides an theoretical analysis on the upper bound of estimation error (i.e. worst case performance). Note that the error bound can be computed from observable quantities. In the experiments, we evaluated the practical utility of this error bound (line 277). To control estimation error, the user can choose a threshold and reject the estimates whose bounds exceed the threshold.
>
> ### (4) Experiments
>
> We did not include SC in the real data study because SC is not able to handle irregularly sampled EHR data. We will clarify this point in the revision.
>
> We will visualize the predicted and true outcomes in the revision to better contextualize the performance gain.
>
> Actually, we have included a propensity weighting benchmark in the experiments: Recurrent Marginal Structural Network (RMSN, line 261). RMSN is an extension of Marginal Structural Models and it employs inverse probability of treatment weighting (IPTW). We will highlight this point in the revision.

---

> ### Author Response · Authors · 2021-08-17
> **Dear Reviwer CDQY**
>
> Once again, thank you for your invaluable feedback. We were wondering whether our response has addressed your concerns. If you have any additional comments, please let us know, we would be happy to address them.

---

### Decision · Program_Chairs · 2021-09-27

**Decision:**

Accept (Poster)

**Comment:**

This paper proposes a new approach to estimating conditional average treatment effects based on longitudinal data. Overall, authors were responsive to criticism raised by reviewers. However, prior to publication two critical revisions should be made. First, explicitly state the key assumption that if the latent confounders are able to predict pretreatment then they will also be able to predict post treatment (this is somewhat implied by proposition 3, but never explicitly stated). Stating this assumption clearly is critical to ensuring that approach is not inappropriately applied in the future. Second, the authors deviate from the norm in this community reporting MAE in lieu of PEHE (or MSE), without providing a justification. In the revision, please include both MAE and MSE results, or provide a clear justification for why MSE results are omitted.